# Distilling the Knowledge in Data Pruning

**Emanuel Ben Baruch**[1] **Adam Botach**[1] **Igor Kviatkovsky**[1] **Manoj Aggarwal**[1] **Gérard Medioni**[1]

## Abstract

With the increasing size of datasets used for training neural networks, data pruning has gained traction in recent years. However, most current data pruning algorithms are limited in their ability to preserve accuracy compared to models trained on the full data, especially in high pruning regimes. In this paper we explore the application of data pruning while incorporating knowledge distillation (KD) when training on a pruned subset. That is, rather than relying solely on ground-truth labels, we also use the soft predictions from a teacher network pre-trained on the complete data. We first establish a theoretical motivation for employing self-distillation to improve training on pruned data. Then, we empirically make a compelling and highly practical observation: using KD, simple random pruning is comparable or superior to sophisticated pruning methods across all pruning regimes. On ImageNet for example, we achieve superior accuracy despite training on a random subset of only 50% of the data. Additionally, we demonstrate a crucial connection between the pruning factor and the optimal knowledge distillation weight. This helps mitigate the impact of samples with noisy labels and low-quality images retained by typical pruning algorithms. Finally, we make an intriguing observation: when using lower pruning fractions, larger teachers lead to accuracy degradation, while surprisingly, employing teachers with a smaller capacity than the student's may improve results.

## 1. Introduction

Recently, data pruning has gained increased interest in the literature due to the growing size of datasets used for training neural networks. Algorithms for data pruning aim to retain the most representative samples of a given dataset and enable the conservation of memory and reduction of computational costs by allowing training on a compact and small subset of the original data. For instance, data pruning can be useful for accelerating hyper-parameter optimization or neural architecture search (NAS) efforts. It may also be used in continual learning or active learning applications.

Existing methods for data pruning have shown remarkable success in achieving good accuracy while retaining only a fraction, $f < 1$, of the original data; see for example (Toneva et al., 2018; Paul et al., 2021; Feldman & Zhang, 2020; Meding et al., 2021) and the overview in (Guo et al., 2022). However, those approaches are still limited in their ability to match the accuracy levels obtained by models trained on the complete dataset, especially in high compression regimes (low $f$).

Score-based data pruning algorithms typically rely on the entire data to train neural networks for selecting the most representative samples. For example, the 'forgetting' method (Toneva et al., 2018) counts for each sample the number of instances during training where the network's prediction for that sample shifts from "correct" to "misclassified". Samples with high rates of forgetting events are assigned higher scores as they are considered harder and more valuable for the training. Other methods use the gradient norm, as in GraNd and EL2N (Paul et al., 2021), or measure changes in the optimal empirical risk, as employed by MoSo (Tan et al., 2023), to score the samples. Typically, once the sample scores are calculated, the models trained on the full dataset are discarded and are no longer in use.

In this paper, we explore the benefit of using a model trained on a complete dataset to enhance training on a pruned subset of the data using knowledge distillation (KD). The motivation behind this approach is that a teacher model trained on the complete dataset captures essential information and core statistics about the entire data. This knowledge can then be utilized when training on a pruned subset. While KD has been extensively studied and demonstrated significant improvements in tasks such as model compression, herein we aim to investigate its impact in the context of data pruning and propose innovative findings for practical usage. Note that, in contrast to traditional model compression techniques, here we focus on self-distillation (SD), where

---

[1]Amazon. Correspondence to: Emanuel Ben Baruch <emanbb@amazon.com>.

*Proceedings of the 42nd International Conference on Machine Learning*, Vancouver, Canada. PMLR 267, 2025. Copyright 2025 by the author(s).

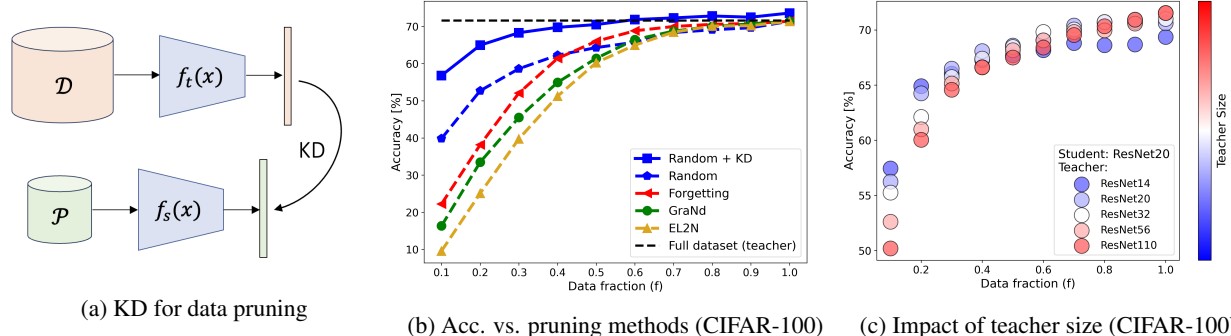

(a) KD for data pruning      (b) Acc. vs. pruning methods (CIFAR-100)      (c) Impact of teacher size (CIFAR-100)

*Figure 1.* **Knowledge distillation for data pruning.** (a) We investigate the usage of a teacher model, pre-trained on a full dataset, to guide a student model during training on a pruned subset of the same data. (b) We find that by integrating KD into the training, simple random pruning outperforms other sophisticated pruning algorithms across all pruning regimes. (c) Interestingly, we observe that when using small data fractions, training with large teachers degrades accuracy, while smaller teachers are favored. This suggests that in high pruning regimes (low $f$), the training is more sensitive to the capacity gap between the teacher and the student.

the teacher and student have identical architectures. The training scheme is illustrated in Figure 1a.

We experimentally demonstrate that incorporating the (soft) predictions provided by the teacher throughout the training process on the pruned data significantly and consistently improves accuracy across multiple datasets, various pruning algorithms, and all pruning fractions (see Figure 1b for example). In particular, using KD, we can achieve comparable or even higher accuracy with only a small portion of the data (e.g., retaining $50\%$ and $10\%$ of the data for CIFAR-100 and SVHN, respectively). Moreover, a dramatic improvement is achieved especially for small pruning fractions (low $f$). For example, on CIFAR-100 with pruning factor $f = 0.1$, accuracy improves by $17\%$ (from $39.8\%$ to $56.8\%$) using random pruning. On ImageNet with $f = 0.1$, the Top-5 accuracy increases by $5\%$ (from $82.37\%$ to $87.19\%$) using random pruning, and by $20\%$ (from $62.47\%$ to $82.47\%$) using EL2N. To explain these improvements, we provide theoretical motivation for integrating SD when training on pruned data. Specifically, we show that using a teacher trained on the entire data reduces the bias of the student's estimation error.

In addition, we present several empirical key observations. First, our results demonstrate that simple random pruning outperforms other sophisticated pruning algorithms in high pruning regimes (low $f$), both with and without knowledge distillation. Notably, prior research demonstrated this phenomenon in the absence of KD (Sorscher et al., 2022; Zheng et al., 2022). Second, we demonstrate a useful connection between the pruning factor $f$ and the optimal weight of the KD loss. Generally, utilizing data pruning algorithms to select high-scoring samples amplifies sensitivity to samples with noisy labels or low quality. This is because keeping the hardest samples increases the portion of these samples as we retain a smaller data fraction. Based on this observation, we

propose to adapt the weight of the KD loss according to the pruning factor. That is, for low pruning factors, we should increase the contribution of the KD term as the teacher's soft predictions reflect possible label ambiguity embedded in the class confidences. On the other hand, when the pruning factor is high, we can decrease the contribution of the KD term to rely more on the ground-truth labels.

Finally, we observe a striking phenomenon when training with KD using larger teachers: in high pruning regimes (low $f$), the optimization becomes more sensitive to the capacity gap between the teacher and the student model. This relates to the well known *capacity gap* problem (Mirzadeh et al., 2019). Interestingly, we find that for small pruning fractions, the student benefits more from teachers with equal or even smaller capacities than its own, see Figure 1c.

The contributions of the paper can be summarized as follows:

- Utilizing KD in data pruning, we find that training is robust to the choice of pruning mechanism at high pruning fractions. Notably, random pruning with KD achieves comparable or superior accuracy compared to other sophisticated methods across all pruning regimes.

- We theoretically show, for the case of linear regression, that using a teacher trained on the entire data reduces the bias of the student's estimation error.

- We demonstrate that by appropriately choosing the KD weight, one can mitigate the impact of label noise and low-quality samples that are retained by common pruning algorithms.

- We make the striking observation that, for small pruning fractions, increasing the teacher size degrades accuracy, while, intriguingly, using teachers with smaller capacities than the student's improves results.

## 2. Related work

**Data pruning.** Data pruning, also known as coreset selection (Mirzasoleiman et al., 2019; Huggins et al., 2016; Tolochinsky & Feldman, 2018), refers to methods aiming to reduce the dataset size for training neural networks. Recent approaches have shown significant progress in retaining less data while maintaining high classification accuracy (Toneva et al., 2018; Paul et al., 2021; Feldman & Zhang, 2020; Meding et al., 2021; Chitta et al., 2019; Sorscher et al., 2022). In (Sorscher et al., 2022), the authors showed theoretically and empirically that data pruning can improve the power law scaling of the dataset size by choosing an optimal pruning fraction as a function of the initial dataset size. Additionally, studies in (Sorscher et al., 2022; Ayed & Hayou, 2023) have demonstrated that existing pruning algorithms often underperform when compared to random pruning methods, especially in high pruning regimes. In (Zheng et al., 2022), the authors suggested a theoretical explanation to this accuracy drop, and proposed a coverage-centric pruning approach which better handles the data coverage. Also, in (Yang et al., 2022), the authors proposed to model the sample selection procedure as a constrained discrete optimization problem.

Recently, several pruning methods have been introduced to address specific limitations of earlier approaches. (Tan et al., 2023) introduced an alternative pruning technique to the costly leave-one-out procedure, leveraging a first-order approximation. This approach assigns higher scores to samples whose gradients consistently align with the gradient expectations across all training stages. D2 (Maharana et al., 2024) proposes a graph-based formulation to represent the data distribution, enabling the selection of a coreset that favors both diverse and difficult regions of the data space. DUAL (Cho et al., 2025) identifies influential training examples early in the learning process, leveraging early signals to guide pruning. In contrast, (Xia et al., 2023) introduce Moderate Coreset, which—given any scoring function—retains the samples whose scores lie near the median, aiming to obtain a lightweight subset that remains robust across diverse data scenarios.

Data pruning proves valuable at reducing memory and computational cost in various applications, including tasks such as hyper-parameter search (Coleman et al., 2019), NAS (Dai et al., 2020), continual and incremental learning (Lange et al., 2019), as well as active learning (Mirzasoleiman et al., 2019; Chitta et al., 2019).

Other related fields are dataset distillation and data-free knowledge distillation (DFKD). Dataset distillation approaches (Wang et al., 2018; Zhao et al., 2020; Yu et al., 2023) aim to compress a given dataset by synthesizing a small number of samples from the original data. The goal of DFKD is to employ model compression in scenarios where the original dataset is inaccessible, for example, due to privacy concerns. Common approaches for DFKD involve generating synthetic samples suitable for KD (Luo et al., 2020; Yoo et al., 2019) or inverting the teacher's information to reconstruct synthetic inputs (Nayak et al., 2019; Yin et al., 2019). Recently, the works in (Cui et al., 2022; Yin et al., 2023), utilized pseudo labels in training with dataset distillation. Unlike dataset distillation and DFKD, which include synthetic data generation, our work focuses on enhancing models trained on pruned datasets created through sample selection, using KD. Moreover, this paper presents practical and innovative findings for applying KD in data pruning.

**Knowledge distillation.** Knowledge distillation is a popular method aiming at distilling the knowledge from one network to another. It is often used to improve the accuracy of a small model using the guidance of a large teacher network (Bucila et al., 2006; Hinton et al., 2015). In recent years, numerous variants and extensions of KD have been developed. For example, (Zagoruyko & Komodakis, 2016; Romero et al., 2014) utilized feature activations from intermediate layers to transfer knowledge across different representation levels. Other methods have proposed variants of KD criteria (Yim et al., 2017; Huang & Wang, 2017; Kim et al., 2018; Ahn et al., 2019), as well as designing objectives for representation distillation, as demonstrated in (Tian et al., 2019; Chen et al., 2020). Self-distillation (SD) refers to the case where the teacher and student have identical architectures. It has been demonstrated that accuracy improvement can be achieved using SD (Furlanello et al., 2018). Recently, theoretical findings were introduced for self-distillation in the presence of label noise (Das & Sanghavi, 2023).

In our paper, we explore the process of distilling knowledge from a model trained on a large dataset to a model trained on a pruned subset of the original data. We focus on self-distillation and present several striking observations that emerge when integrating SD for data pruning.

## 3. Method

Given a dataset $\mathcal{D}$ with $N$ labeled samples $\{x_i, y_i\}_{i=1}^N$, a data pruning algorithm $\mathcal{A}$ aims at selecting a subset $\mathcal{P} \subset \mathcal{D}$ of the most representative samples for training. We denote by $f$ the pruning factor, which represents the fraction of data to retain, calculated as $f = N_f/N$ where $N_f$ is the size of the pruned dataset. Note that $0 < f < 1$. Score-based algorithms assign a score to each sample, representing its importance in the learning process. Let $s_i$ be the score corresponding to a sample $x_i$, sorting them in a descending order $s_{k_1} > s_{k_2}, ..., > s_{k_N}$, following the sorting indices $\{k_1, ..., k_N\}$, we obtain the pruned dataset by retaining the highest scoring samples, $\mathcal{P} = \{x_{k_1}, ..., x_{k_{N_f}}\}$. Usually, score-based algorithms retain hard samples while excluding the easy ones. Note that in random pruning, we simply sample the indices $k_1, ..., k_N$ uniformly. In this paper, given

a pruning algorithm $\mathcal{A}$, our objective is to train a model on the pruned dataset $\mathcal{P}$ while maximizing accuracy.

## 3.1. Training on the pruned dataset using KD

Typically, score-based pruning methods involve training multiple models on the full dataset $\mathcal{D}$ to compute the scores (Toneva et al., 2018; Paul et al., 2021; Feldman & Zhang, 2020; Meding et al., 2021). These models are discarded and are not utilized further after the scores are computed. We argue that a model trained on the full dataset encapsulates valuable information about the entire distribution of the data and its classification boundaries, which can be leveraged when training on the pruned data $\mathcal{P}$. In this work, we investigate a training scheme which incorporates the soft predictions of a teacher network, pre-trained on the full dataset, throughout training on the pruned data.

Let $f_t(x)$ be the teacher backbone pre-trained on $\mathcal{D}$. The teacher outputs logits $\{z_i\}_{i=1}^C$, where $C$ is the number of classes. The teacher's soft predictions are computed by,

$$q_i = \frac{\exp(z_i/\tau)}{\sum_j \exp(z_j/\tau)}, \quad i = 1 \ldots C, \tag{1}$$

where $\tau$ is the temperature hyper-parameter. Similarly, we denote the student model trained on the dataset $\mathcal{P}$ as $f_s(x; \theta)$, where $\theta$ represents the student's parameters. The student's $i$-th soft prediction is denoted by $p_i(\theta)$. We optimize the student model using the following loss function,

$$\mathcal{L}(\theta) = (1 - \alpha)\mathcal{L}_{\text{cls}}(\theta) + \alpha\mathcal{L}_{\text{KD}}(\theta), \tag{2}$$

where the classification loss $\mathcal{L}_{\text{cls}}(\theta)$ measures the cross-entropy between the ground-truth labels and the student's predictions, represented as: $-\sum_i y_i \log p_i(\theta)$. For the KD term $\mathcal{L}_{\text{KD}}(\theta)$, a common choice is the Kullback-Leibler (KL) divergence between the soft predictions of the teacher and the student. The hyper-parameter $\alpha$ controls the weight of the KD term relative to the classification loss.

Integrating the KD loss into the training process allows us to leverage the valuable knowledge embedded in the teacher's soft predictions $q_i$. These predictions may encapsulate potential relationships between categories and class hierarchies, accumulated by the teacher during its training on the entire dataset. Intuitively, reliable data and class distributions can be effectively learned from large datasets, but are harder to infer from small datasets.

In Section 4.1, we empirically demonstrate that integrating knowledge distillation into the optimization process of the student model, trained on pruned data, leads to significant improvements across all pruning factors and various pruning methods. In addition, we show that simple random pruning outperforms other sophisticated pruning methods for low pruning fractions (low $f$), both with and without knowledge

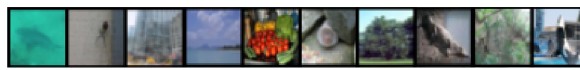

(a) CIFAR-100 highest score pruning samples

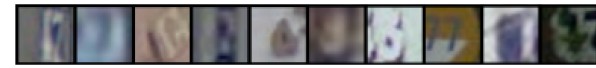

(b) SVHN highest score pruning samples

*Figure 2.* **Highest scoring samples.** Top 10 highest scoring samples selected by the 'forgetting' pruning method for CIFAR-100 and SVHN datasets. The labels of the majority of the images are ambiguous due to class complexity or low image quality.

distillation. We note that prior work has demonstrated this phenomenon in the absence of KD (Sorscher et al., 2022). Interestingly, we also observe that training with KD is robust to the choice of the data pruning method, including simple random pruning, for sufficiently high pruning fractions.

These observations on the effectiveness of random pruning in the presence of KD are compelling, especially in scenarios where data pruning occurs unintentionally as a by-product of the system, such as cases where the full dataset is no longer accessible due to privacy concerns. However, using knowledge distillation we can train a student model on the remaining available data while maintaining a high level of accuracy.

## 3.2. Mitigating noisy samples in pruned datasets

In general, hard samples are essential for the optimization process as they are located close to the classification boundaries. However, retaining the hardest samples while excluding moderate and easy ones increases the proportion of samples with noisy and ambiguous labels, or images with poor quality. For example, in Figure 2, we present the highest scoring images selected by the 'forgetting' pruning algorithm for CIFAR-100 and SVHN. As can be seen, in the majority of the images determining the class is non-trivial due to the complexity of the category (e.g., fine-grained classes) or due to poor quality. By using knowledge distillation, the student can learn such label ambiguity and mitigate noisy labels.

In a recent work (Das & Sanghavi, 2023) it was demonstrated that the benefit of using a teacher's predictions increases with the degree of label noise. Consequently, it was found that more weight should be assigned to the KD term as the noise variance increases. Similarly, in our work we empirically demonstrate that as the pruning factor $f$ becomes lower, we should rely more on the teacher's predictions by increasing $\alpha$ in Eq. 2. Conversely, as the pruning factor is increased, we may rely more on the ground-truth labels by decreasing $\alpha$. We find that setting $\alpha$ properly is crucial when applying pruning methods that retain hard sam-

ples. Formally, the objective should be aware of the pruning fraction $f$ as follows,

$$\mathcal{L}(\theta, f) = (1 - \alpha(f))\mathcal{L}_{\text{cls}}(\theta) + \alpha(f)\mathcal{L}_{\text{KD}}(\theta). \quad (3)$$

For example, as can be seen from Figure 6, when the pruning fraction is low ($f = 0.1$), training with $\alpha = 1$ is superior, achieving more than $8\%$ higher accuracy compared to $\alpha = 0.5$. Conversely, for high pruning fractions (e.g. $f = 0.7$), using $\alpha = 0.5$ outperforms $\alpha = 1$ by more than $1\%$ accuracy. We further explore the relationship between $\alpha$ and $f$ in Section 4.3.

### 3.3. Theoretical motivation

In this section we provide a theoretical motivation for the success of self-distillation in enhancing training on pruned data. We base our analysis on the recent results reported in (Das & Sanghavi, 2023) for the case of regularized linear regression. Note that while we use logistic regression in practice, we anchor our theoretical results in linear regression for the sake of simplicity. Also, it often allows for a reliable emulation of outcomes observed in processes applied to logistic regression (see e.g. in (Das & Sanghavi, 2023)). In particular, we show that employing self-distillation using a teacher model trained on a larger dataset reduces the error bias of the student estimation.

We are given a data matrix, $\mathbf{X} = [\mathbf{x}_1, \ldots, \mathbf{x}_N] \in \mathbb{R}^{d \times N}$, and a corresponding label vector $\mathbf{y} = [y_1, \ldots, y_N] \in \mathbb{R}^N$, where $N$ and $d$ are the number of samples and their dimension, respectively. Let $\boldsymbol{\theta}^* \in \mathbb{R}^d$ be the ground-truth model parameters. The labels are assumed to be random variables, linearly modeled by $\mathbf{y} = \mathbf{X}^T \boldsymbol{\theta}^* + \boldsymbol{\eta}$, where $\boldsymbol{\eta} \in \mathbb{R}^N$ is assumed to be Gaussian noise, uncorrelated and independent on the observations. In data pruning, we select $N_f$ columns from $\mathbf{X}$ and their corresponding labels: $\mathbf{X}_f \in \mathbb{R}^{d \times N_f}$, $\mathbf{y}_f \in \mathbb{R}^{N_f}$. Thus, $\mathbf{y}_f = \mathbf{X}_f^T \boldsymbol{\theta}^* + \boldsymbol{\eta}_f$. We also assume that $d \leq N_f \leq N$ which is true in most practical scenarios. Solving linear regularized regression using pruned dataset with fraction $f$, the parameters of the trained model are obtained by:

$$\hat{\boldsymbol{\theta}}(f) = \underset{\boldsymbol{\theta}}{\arg\min} \left\{ ||\mathbf{y}_f - \mathbf{X}_f^T \boldsymbol{\theta}||_2^2 + \frac{\lambda}{2} ||\boldsymbol{\theta}||_2^2 \right\}$$
$$= (\mathbf{X}_f \mathbf{X}_f^T + \lambda \mathbf{I}_d)^{-1} \mathbf{X}_f \mathbf{y}_f,$$

where $\lambda > 0$ is the regularization hyper-parameter, and $\mathbf{I}_d \in \mathbb{R}^{d \times d}$ is the identity matrix. Note that a teacher trained on the full data is given by: $\hat{\boldsymbol{\theta}}_t = \hat{\boldsymbol{\theta}}(1) = (\mathbf{X}\mathbf{X}^T + \lambda \mathbf{I}_d)^{-1} \mathbf{X}\mathbf{y}$.

Here, we look at the more general case where the student is trained on a pruned subset with factor $f$, and the teacher model is trained on a larger subset of the data, $f_t > f$. Following (Das & Sanghavi, 2023), the model learned by the student is given by,

$$\hat{\boldsymbol{\theta}}_s(\alpha, f, f_t) = (1 - \alpha)(\mathbf{X}_f \mathbf{X}_f^T + \lambda \mathbf{I}_d)^{-1} \mathbf{X}_f \mathbf{y}_f$$
$$+ \alpha(\mathbf{X}_f \mathbf{X}_f^T + \lambda \mathbf{I}_d)^{-1} \mathbf{X}_f \hat{\mathbf{y}}_f^{(t)} \quad (4)$$
$$= (\mathbf{X}_f \mathbf{X}_f^T + \lambda \mathbf{I}_d)^{-1} \mathbf{X}_f ((1 - \alpha)\mathbf{y}_f + \alpha \mathbf{X}_f^T \hat{\boldsymbol{\theta}}(f_t)),$$

where $\hat{\mathbf{y}}_f^{(t)} = \mathbf{X}_f^T \hat{\boldsymbol{\theta}}(f_t)$, i.e., , the teacher's predictions of the student's samples $\mathbf{X}_f$. Note that in a regular self-distillation (without pruning), we have $f = f_t = 1$, and $\alpha > 0$. Also, in a regular training on pruned data (without KD), $f < 1$, and $\alpha = 0$. In our scenario we utilize self-distillation for data pruning, i.e., , $f < f_t \leq 1$, and $\alpha > 0$.

We denote the student estimation error as $\boldsymbol{\epsilon}_s(\alpha, f, f_t) = \hat{\boldsymbol{\theta}}_s(\alpha, f, f_t) - \boldsymbol{\theta}^*$. In (Das & Sanghavi, 2023), the authors show that employing self-distillation ($\alpha > 0$) reduces the variance of the student estimation, but on the other hand, increases its bias. In the following, we show that distilling the knowledge from a teacher trained on a larger data subset w.r.t the student, decreases the error estimation bias.

**Theorem 3.1.** *Let $\mathbf{X} \in \mathbb{R}^{d \times N}$ and $\mathbf{y} \in \mathbb{R}^N$ be the full observation matrix and label vector, respectively. Let $\mathbf{y}_f = \mathbf{X}_f^T \boldsymbol{\theta}^* + \boldsymbol{\eta}_f$, where $\boldsymbol{\theta}^*$ is the ground-truth projection vector and $\boldsymbol{\eta}_f \in \mathbb{R}^N$ is a Gaussian uncorrelated noise independent on $\mathbf{X}$. Let $\boldsymbol{\epsilon}_s(\alpha, f, f_t) = \hat{\boldsymbol{\theta}}_s(\alpha, f, f_t) - \boldsymbol{\theta}^*$ be the student estimation error. Also, assume that $d \leq N_f \leq N$, and $f \leq f_t$. Then, for any $\alpha$,*

$$||\mathbb{E}_\eta[\boldsymbol{\epsilon}_s(\alpha, f, f_t)]||^2 \leq ||\mathbb{E}_\eta[\boldsymbol{\epsilon}_s(\alpha, f, f))]||^2.$$

We include the proof for Theorem 3.1 in the supplementary. As data pruning is susceptible to label noise due to retaining the hardest samples, this finding demonstrates the utility of the proposed method. It suggests that employing self-distillation with a teacher trained on the entire dataset ($f_t = 1$) enables the reduction of estimation bias in a student trained on a pruned subset. In Section 4.2 we analyze the impact of different $f_t$ values on the student's accuracy, with the corresponding results illustrated in Figure 5.

## 4. Experimental results

In this section we provide empirical evidence for our method through extensive experimentation over a variety of datasets, an assortment of data pruning methods and across a wide range of pruning levels. Then, we also investigate how the KD weight, the teacher size and the KD method affect student performance under different pruning regimes.

**Datasets**. We perform experiments on four classification datasets: CIFAR-10 (Krizhevsky et al., a) with 10 classes, consists of 50,000 training samples and 10,000 testing samples; SVHN (Netzer et al., 2011) with 10 classes, consists of 73,257 training samples and 26,032 testing samples; CIFAR-100 (Krizhevsky et al., b) with 100 classes, consists of 50,000 training samples and 10,000 testing samples;

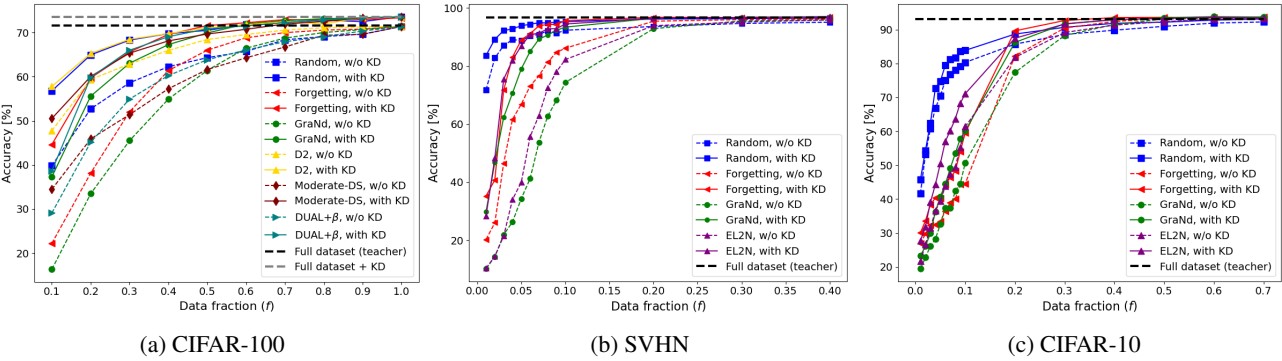

(a) CIFAR-100          (b) SVHN          (c) CIFAR-10

*Figure 3.* **Data pruning results with knowledge distillation.** Accuracy results across different pruning factors $f$, and various pruning approaches on the CIFAR-100, SVHN, and CIFAR-10 datasets. We use an equalized weight in the loss (i.e., $\alpha = 0.5$). Using KD, significant improvement is achieved across all pruning regimes and all pruning methods. Random pruning outperforms other pruning methods for low pruning factors. For sufficiently high $f$, the accuracy is robust to the choice of the pruning approach in the presence of knowledge distillation.

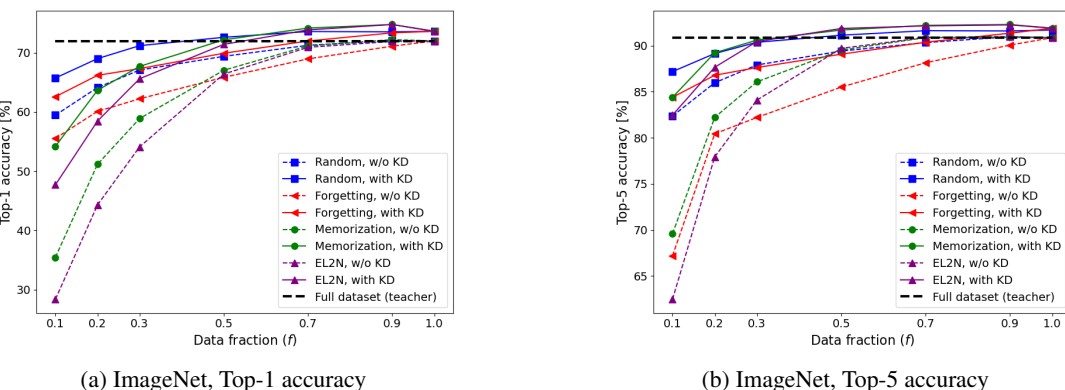

(a) ImageNet, Top-1 accuracy          (b) ImageNet, Top-5 accuracy

*Figure 4.* **Data pruning results with KD on ImageNet.** Accuracy results across different pruning factors $f$, and various pruning methods on the ImageNet dataset. We use an equalized weight ($\alpha = 0.5$) in Eq. 2.

and ImageNet (Russakovsky et al., 2015) with 1,000 classes, consists of 1.2M training samples and 50K testing samples.

**Pruning Methods**. We utilize several data-pruning algorithms: 'forgetting' (Toneva et al., 2018), Gradient Norm (GraNd), Error L2-Norm (EL2N) (Paul et al., 2021), 'memorization'[1] (Feldman & Zhang, 2020), D2 (Maharana et al., 2024), DUAL (Cho et al., 2025), and Moderate-coreset (Xia et al., 2023). We also utilize a class-balanced random pruning scheme, which, given a pruning budget, randomly and equally draws samples from each class.

### 4.1. Training on pruned data with KD

To demonstrate the advantage of incorporating KD-based supervision when training on pruned data, we utilize the aforementioned data pruning methods on each dataset using a wide range of pruning factors. Then, we train models on the produced data subsets with and without KD. We note

---
[1]We note that while the authors of *memorization* did not originally utilize the method for data pruning, its efficacy on ImageNet was later demonstrated by (Sorscher et al., 2022).

that in the presence of KD the respective teachers that are utilized are trained on the full datasets.

As can be observed in Figures 3 and 4, the incorporation of KD into the training process consistently enhances model accuracy across all of the tested scenarios, regardless of the tested dataset, pruning method or pruning level. For example, compared to baseline models trained on the full datasets without KD, utilizing KD can lead to comparable accuracy levels by retaining only small portions of the original datasets (e.g., 10%, 30%, 50% on SVHN, CIFAR-10, and CIFAR-100, respectively, using 'forgetting'). In fact, even on a large scale dataset as ImageNet, comparable accuracy can be achieved by randomly retaining just 30% of the data, while training on larger subsets remarkably results in superior accuracy to the baseline (e.g., +1.6% using a random subset of 70%).

As shown in Figure 3, at low pruning fractions, random pruning combined with KD consistently outperforms all other methods, with the exception of D2 + KD, which achieves comparable performance. However, unlike D2—which re-

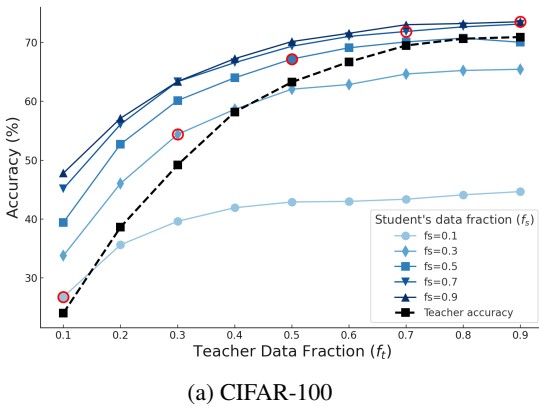

(a) CIFAR-100

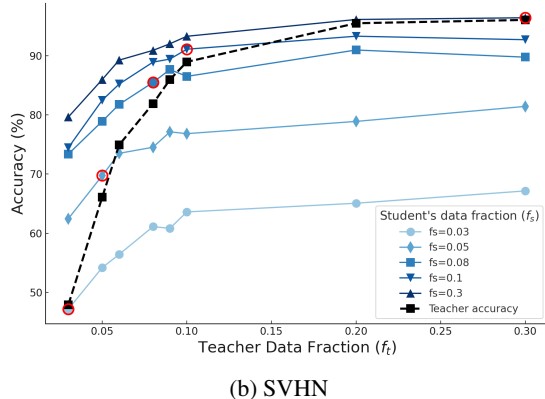

(b) SVHN

*Figure 5.* **Accuracy versus teacher data fraction** ($f_t$). The parameters $f_s$ and $f_t$ represent the fractions of data used to train the student and teacher models, respectively. The red circles emphasize the self-distillation (SD) accuracy, while the dashed black line depicts the teacher's accuracy. This figure highlights two insights: (1) increasing $f_t$ consistently improves accuracy on top of self-distillation; (2) in all scenarios, SD outperforms standard training without knowledge distillation, as indicated by the circles being positioned above the dashed purple curve. These results support the theoretical motivation presented in Section 3.3.

quires careful tuning of multiple hyperparameters ($k, \beta, \gamma_r$) for each pruning fraction and dataset—our approach, based on simple random pruning with KD, involves no such tuning. This makes it significantly more practical and easier to apply in real-world scenarios.

Moreover, we note that the accuracy gains due to KD are most significant in high-compression scenarios. For instance, on CIFAR-100 with $f = 0.1$, KD contributes to absolute accuracy improvements of 17%, 22.4%, 21%, and 19.7% across the random, 'forgetting', GraNd, and EL2N pruning methods, respectively. Similarly, on SVHN, which permits even stronger compression, improvements of the same order of magnitude can be observed at a lower pruning factor ($f = 0.01$).

These findings support the idea that the soft-predictions produced by a well-informed teacher contain rich and valuable information that can greatly benefit a student in a limited-data setting. This 'dark knowledge', notably absent in conventional one-hot labels, allows the student to deduce stronger generalizations from each available data sample, which in turn translates to better performance given the same training data.

Finally, two additional interesting patterns emerge from our experiments. First, in high-compression scenarios (e.g., $f \leq 0.4$ in CIFAR-100, $f \leq 0.08$ in SVHN), it is evident that random pruning surpasses all other methods in effectiveness, both with and without KD. This aligns with the notion that aggressive pruning via score-based techniques retains larger concentrations of low quality or noisy samples due to mistaking them for challenging cases. This phenomenon was previously noted without KD in (Sorscher et al., 2022). Second, under low-compression conditions (e.g., $f \geq 0.5$ in CIFAR-100, $f \geq 0.2$ in SVHN), we observe that KD

renders the student model robust to the pruning technique used. This finding is significant as it suggests that it may be possible to forgo state-of-the-art pruning techniques in favor of basic random pruning in the presence of KD.

### 4.2. Impact of Teacher's Training Data Fraction

Up to this point, we employed a teacher trained on the full dataset, i.e., $f_t = 1$. We now explore how training the teacher on smaller data fractions ($0 < f_t < 1$) affects the student's accuracy. Figure 5 presents the student's accuracy on CIFAR-100 and SVHN across different data fractions used to train the teacher and the student. The results highlight two key findings: (1) increasing $f_t$ consistently enhances accuracy beyond SD; (2) in every scenario, SD surpasses standard training without KD. These observations align with the theoretical insights discussed in Section 3.3.

### 4.3. Adapting the KD weight vs. the pruning factor

We wish to investigate how varying the KD weight $\alpha$ affects the performance of the student under different pruning levels of a given dataset. To explore this we conduct experiments on CIFAR-100 with 'forgetting' as the pruning method and present the results in Figure 6. As can be observed, lower pruning fractions favor higher values of $\alpha$, while higher pruning fractions advocate for lower ones. As explained earlier, aggressive pruning via score-based methods tends to result in subsets with greater proportions of label noise and low quality samples. Hence, for lower pruning factors, increasing the KD weight seems to help the student mitigate the extra noise by relying more on the teacher's predictions. Conversely, as the pruning factor increases and the proportions of noise in the pruned subset gradually diminish, it appears to be beneficial for the student to balance the contributions of KD and the ground-truth labels. Similar results

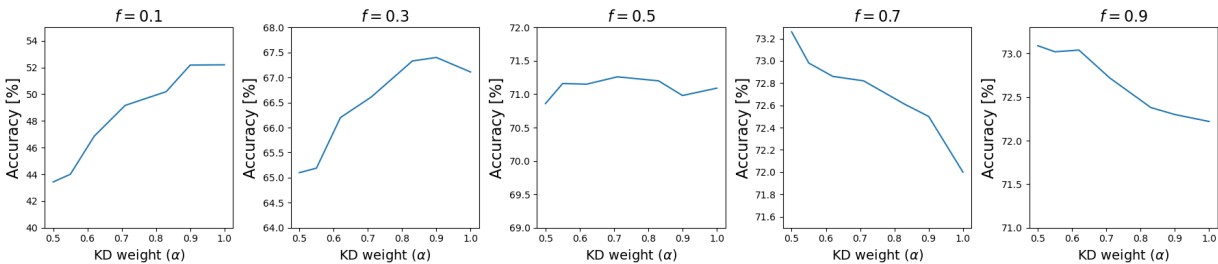

*Figure 6.* **Optimal KD weight versus pruning factor.** Accuracy is presented for CIFAR-100 while varying the KD weight $\alpha$ for different pruning factors. We utilize 'forgetting' as the pruning method. For low pruning fractions (low $f$), accuracy generally increases when increasing the KD weight to rely more on the teacher's soft predictions. As we use higher pruning fractions (high $f$), it is usually better to lower $\alpha$ in order to increase the contribution of the ground-truth labels.

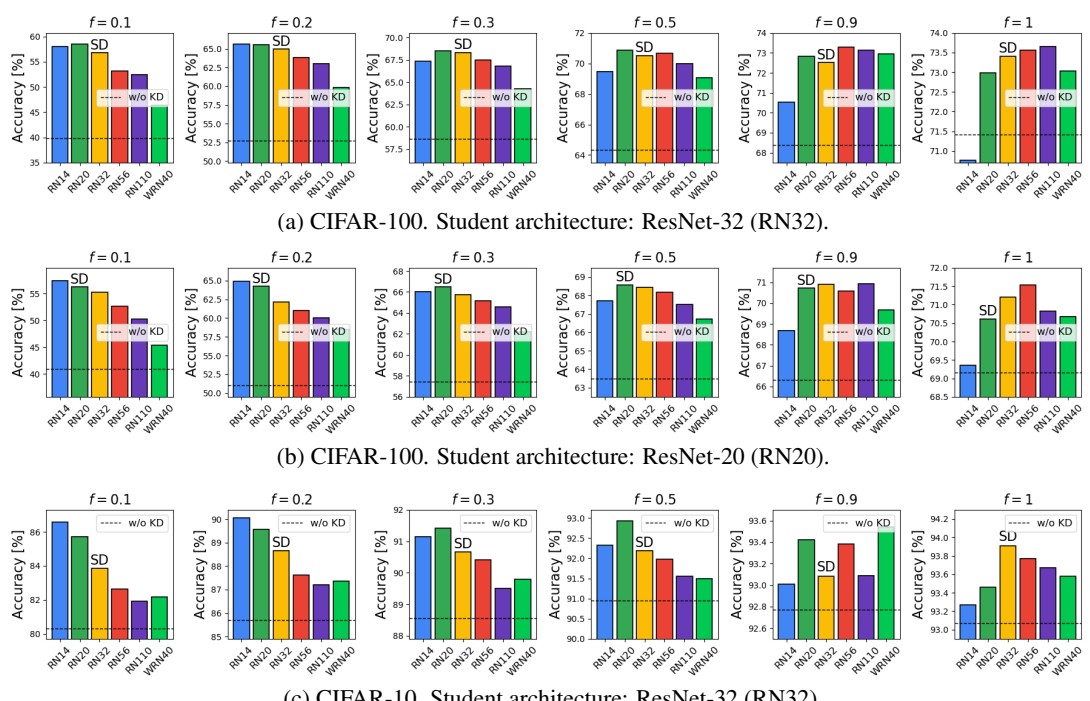

(a) CIFAR-100. Student architecture: ResNet-32 (RN32).

(b) CIFAR-100. Student architecture: ResNet-20 (RN20).

(c) CIFAR-10. Student architecture: ResNet-32 (RN32).

*Figure 7.* **Exploring the effect of the teacher's capacity.** Accuracy results for a student with (a) ResNet-32 and (b) ResNet-20 architectures while using teacher models with increasing capacities along the horizontal axes. In each instance, we denote the teacher whose architecture matches that of the student by 'SD' (self-distillation). We use random pruning with different fractions. Interestingly, under low pruning factors, increasing the teacher's capacity results in lower student accuracy.

on SVHN can be found in the supplementary.

## 4.4. Using teachers of different capacities

Until now, we have focused on the case where both the student and teacher share the same architecture (i.e., self-distillation). In this section, we explore how the capacity of the teacher affects the student's performance across different pruning regimes. In Figure 7a, we present accuracy results across various pruning factors for the case of randomly pruning CIFAR-100 and training with a ResNet-32 student. We employ 6 teacher architectures of increasing ca-

pacities: (1) ResNet-14 with $69.9\%$ accuracy, (2) ResNet-20 with $70.23\%$ accuracy, (3) ResNet-32 with $71.6\%$ accuracy, (4) ResNet-56 with $72.7\%$ accuracy, (5) ResNet-110 with $74.4\%$ accuracy, and (6) WRN-40-2 with $75.9\%$ accuracy. Also, note that for each teacher architecture we experiment with five different temperature values in the range $2 - 7$. We show the impact of the temperature selection in the supplementary. Similarly, in Figure 7b we present results for the same experiment using a ResNet-20 student, while Figure 7c depicts results of a similar experiment on CIFAR-10 for the ResNet-32 student. As observed, at low pruning factors, increasing the teacher's capacity harms the accuracy

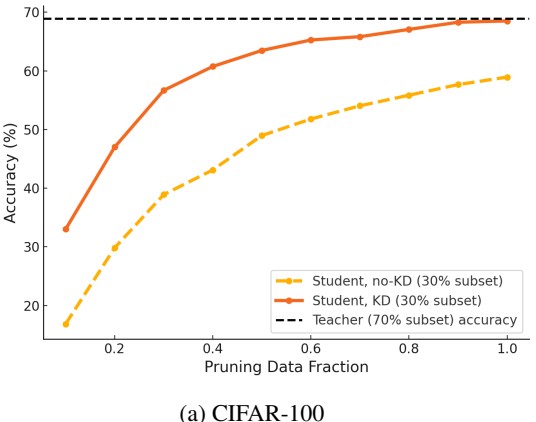

(a) CIFAR-100

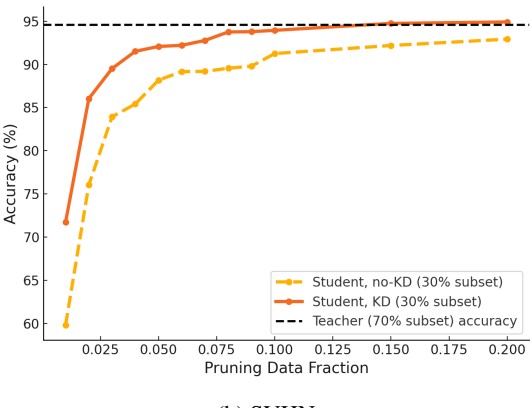

(b) SVHN

*Figure 8.* **Accuracy when the student and teacher are trained on disjoint subsets.** Notably, combining knowledge distillation with data pruning yields significant performance gains, even when the student is trained on a pruned dataset that differs from the teacher's training data. For instance, in CIFAR-100 with random pruning at $f = 50\%$, we observe a 14.5-point accuracy improvement when the teacher model was trained on a different subset.

of the student. This trend is consistently observed across various student architectures and datasets, and is robust to the selection of the KD temperature. Additional results are provided in the supplementary.

This observation highlights a striking phenomenon: the capacity gap problem, which denotes the disparity in architecture size between the teacher and student, becomes more pronounced when applying knowledge distillation during training on pruned data.

### 4.5. Data Pruning in Disjoint Datasets

In this section, we evaluate a practical setting where the pruned dataset is not a subset of the data originally used to train the teacher model. This scenario is highly relevant in real-world applications, particularly in cases where access to the full dataset is restricted, such as due to privacy regulations, as discussed at the end of Section 3.1.

Let $\mathcal{P}$ be a pruned dataset sampled from $\mathcal{D}$ to train the student model, and let $\mathcal{S}$ be the training data for the teacher. In the following experiments, $\mathcal{D}$ and $\mathcal{S}$ and are disjoint i.e., $\mathcal{P} \cap \mathcal{S} = \emptyset$. For the empirical study, we used 70% of the training data to train the teacher and the remaining 30% to train the student with different pruning ratios. Specifically, we compared the performance with and without KD for CIFAR-100 and SVHN datasets.

The experimental results are shown in Figure 8. Notably, combining knowledge distillation with data pruning yields significant performance gains, even when the student is trained on a pruned dataset that differs from the teacher's training data. For instance, in CIFAR-100 with random pruning at $f = 50\%$, we observe a 14.5-point accuracy improvement when the teacher model was trained on a different subset.

## 5. Conclusion

In this paper, we investigated the application of knowledge distillation for training models on pruned data. We demonstrated the significant benefits of incorporating the teacher's soft predictions into the training of the student across all pruning fractions, various pruning algorithms and multiple datasets. We empirically found that incorporating KD while using simple random pruning can achieve comparable or superior accuracy compared to sophisticated pruning approaches. We also demonstrated a useful connection between the pruning factor and the KD weight, and propose to adapt $\alpha$ accordingly. Finally, for small pruning fractions, we made the surprising observation that the student benefits more from teachers with equal or even smaller capacities than that of its own, over teachers with larger capacities.

## Impact Statement

This paper presents work whose goal is to advance the field of Machine Learning. There are many potential societal consequences of our work, none which we feel must be specifically highlighted here.

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

# Appendix

## A. Implementation Details

For computational efficiency we conduct our self-distillation experiments on all datasets using the ResNet-32 (He et al., 2016) architecture, except for ImageNet for which we utilize the larger ResNet-50. Our training and distillation recipes are simple. We utilize SGD with Momentum to optimize the models and incorporate basic data-augmentations during training. Additional implementation details can be found in the supplementary.

### A.1. Obtaining the pruning scores

We utilize the default pruning recipes offered by the DeepCore framework (Guo et al., 2022) in order to compute most of the pruning scores used in our experiments. For SVHN (Netzer et al., 2011), CIFAR-10 (Krizhevsky et al., a) and CIFAR-100 (Krizhevsky et al., b) we compute the scores using the ResNet-34 (He et al., 2016) architecture. For ImageNet (Russakovsky et al., 2015) we compute the scores for the 'forgetting' pruning method (Toneva et al., 2018) using ResNet-50, while for the 'memorization' (Feldman & Zhang, 2020) and EL2N (Paul et al., 2021) methods we directly utilize the scores released by (Sorscher et al., 2022). Specifically, we note that for EL2N on ImageNet we adopt the released variant of the scores which was averaged over 20 models.

### A.2. Conducting the distillation experiments

We conduct our knowledge distillation experiments on the pruned SVHN, CIFAR-10 and CIFAR-100 datasets using a modified version of the RepDistiller framework (Tian et al., 2019). For the most part we adopt the default training and distillation recipes offered by the framework. The models are trained for 240 epochs with a batch size of 64. For the optimization process we use SGD with learning rate 0.05, momentum value of 0.9 and weight decay of $5e^{-4}$. The learning rate is decreased by a factor of 10 on the 150th, 180th and 210th epochs. To conduct the distillation experiments on ImageNet we expand the DeepCore (Guo et al., 2022) framework to support knowledge distillation on pruned datasets. Apart from this change we mostly rely on the default training recipe offered by the framework. The models are trained for 240 epochs with a batch size of 128. We utilize SGD with learning rate 0.1, momentum value of 0.9 and weight decay of $5e^{-4}$. The learning rate is gradually decayed during training using a cosine-annealing scheduler (Loshchilov & Hutter, 2017). In all of our distillation experiments we use $\tau = 4$ as the temperature for the KD's soft predictions computation in Equation (1).

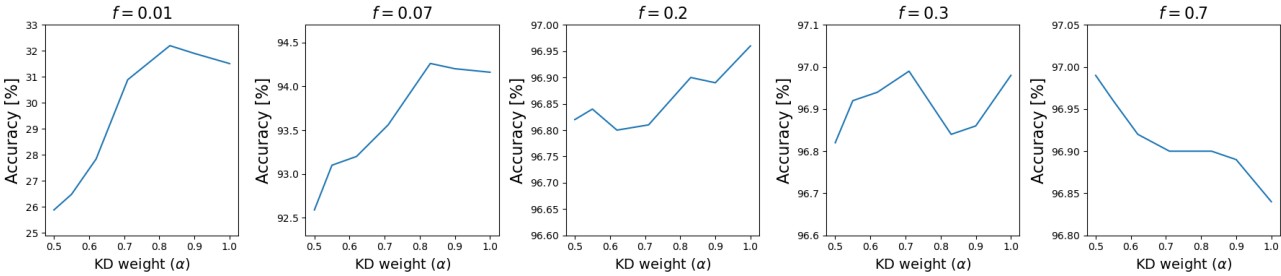

*Figure 9.* **Optimal KD weight versus pruning factor.** Accuracy is presented on SVHN while varying the KD weight $\alpha$ across different pruning factors. We utilize 'forgetting' as the pruning method. For low pruning fractions (low $f$), accuracy generally increases when increasing the KD weight to rely more on the teacher's soft predictions. However, as we use higher pruning fractions (high $f$), it is usually better to use lower $\alpha$ values in order to increase the contribution of the ground-truth labels.

## B. Adapting the KD weight vs. the pruning factor

Following Section 4.3, in Figure 9 we present additional accuracy results which show the effect of varying the KD weight $\alpha$ across different pruning factors $f$, this time on the SVHN dataset. We utilize 'forgetting' as the pruning method. Here, a similar trend to the one previously observed on CIFAR-100 can be seen: for low pruning fractions, accuracy improves as we increase the KD weight, while for higher pruning fractions it is usually better to use lower $\alpha$ values.

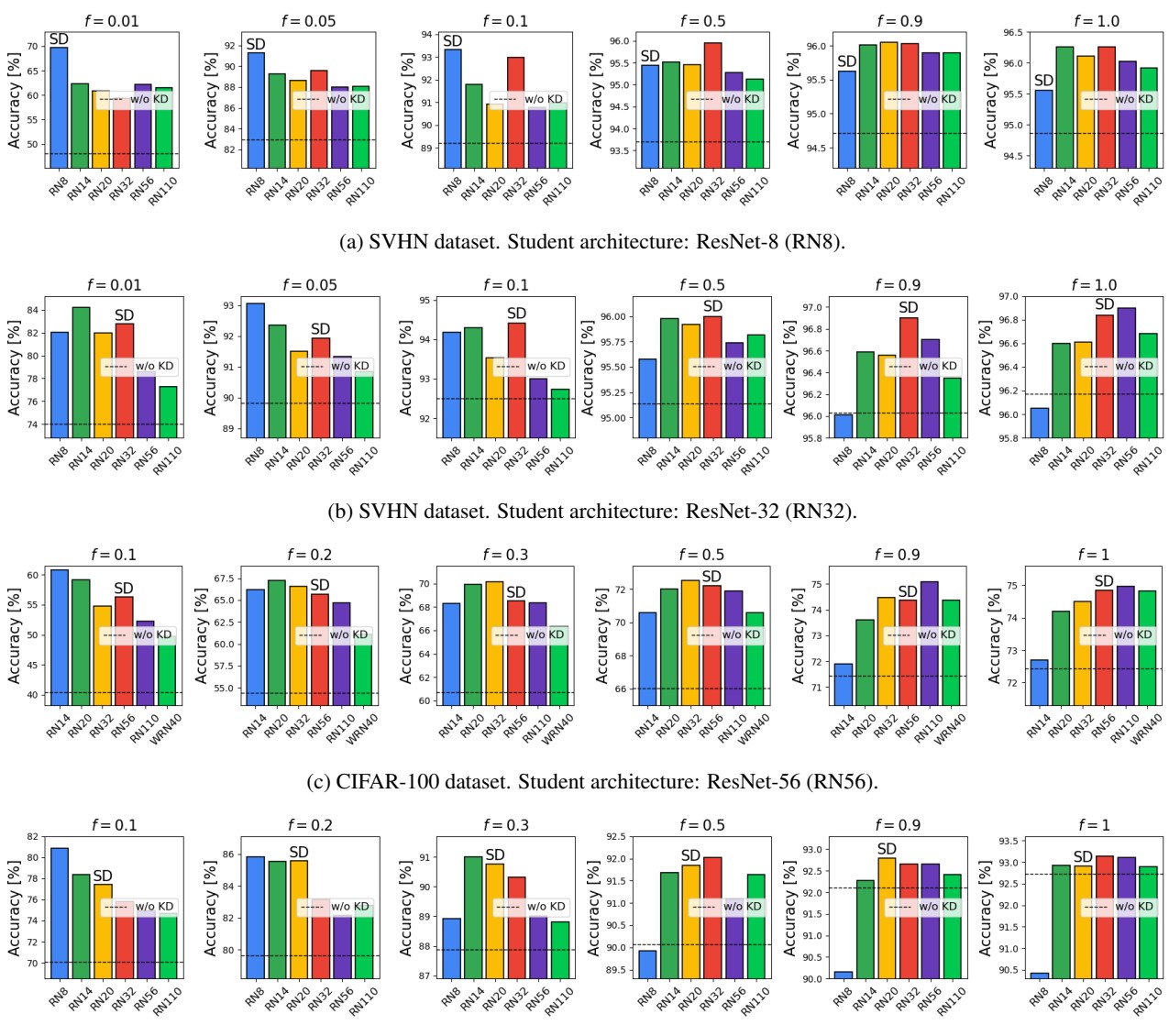

(a) SVHN dataset. Student architecture: ResNet-8 (RN8).

(b) SVHN dataset. Student architecture: ResNet-32 (RN32).

(c) CIFAR-100 dataset. Student architecture: ResNet-56 (RN56).

(d) CIFAR-10 dataset. Student architecture: ResNet-20 (RN20).

*Figure 10.* **Exploring the effect of the teacher's capacity.** Accuracy results across different pruning fractions using teacher models with increasing capacities for: (a) a ResNet-8 student on SVHN, (b) a ResNet-32 student on SVHN, (c) a ResNet-56 student on CIFAR-100, and for (d) a ResNet-20 student on CIFAR-10. Random pruning is utilized. These results further corroborate our observation that teachers with smaller capacities lead to higher student accuracy when utilizing low pruning fractions.

## C. Using teachers of different capacities

In Section 4.4 we have made the observation that teachers with smaller capacities lead to higher student accuracy when utilizing low pruning fractions. Here we provide additional results which demonstrate the consistency of this observation. In Figures 10a and 10b we present student accuracy results on SVHN using different teachers and various pruning fractions, where the utilized student architectures are ResNet-8 and ResNet-32, respectively. Similarly, Figure 10c depicts results on CIFAR-100 with a ResNet-56 student, and Figure 10d shows the same on CIFAR-10 with a ResNet-20 student. Random pruning is utilized in all experiments.

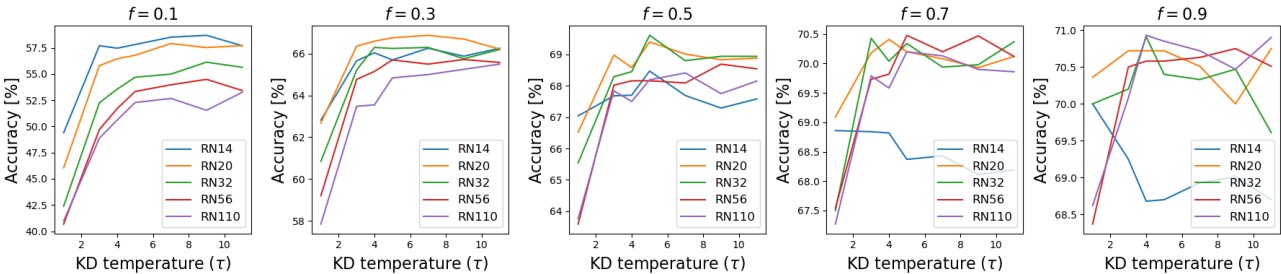

*Figure 11.* **Impact of the KD temperature on the student's accuracy using teachers with different capacities.** We present accuracy results across different pruning fractions on CIFAR-100 for a ResNet-20 student. Random pruning is utilized. As can be seen, for lower pruning fractions (e.g. $f = 0.1$ and $f = 0.3$), teachers with lower capacities outperform teachers with higher capacities.

| Method | 5% | 10% | 30% | 50% |
|---|---|---|---|---|
| w/o KD | 14.46 | 22.21 | 49.41 | 67.47 |
| KD (Hinton et al., 2015) | 28.62 | 46.27 | 66.82 | 70.95 |
| FitNets (Romero et al., 2014) | 25.66 | 44.84 | 65.7 | 70.77 |
| AB (Heo et al., 2018) | 30.5 | 47.68 | 66.15 | 71.22 |
| AT (Zagoruyko & Komodakis, 2016) | 28.26 | 42.59 | 65.75 | 70.45 |
| FT (Kim et al., 2018) | 28.34 | 44.01 | 64.95 | 70.75 |
| FSP (Yim et al., 2017) | 27.62 | 37.16 | 62.79 | 69.72 |
| NST (Huang & Wang, 2017) | 26.2 | 44.5 | 64.93 | 70.97 |
| PKT (Passalis & Tefas, 2018) | 27.3 | 44.09 | 65.22 | 70.7 |
| RKD (Park et al., 2019) | 21.69 | 43.03 | 65.43 | 70.36 |
| SP (Tung & Mori, 2019) | 29.09 | 42.53 | 65.62 | 70.72 |
| VID (Ahn et al., 2019) | **32.5** | **49.46** | **67.38** | **71.16** |

*Table 1.* **Comparison of different KD approaches on several pruning levels of CIFAR-100.** We add various KD loss terms to Eq. 2, in addition to the vanilla KD term. 'Forgetting' is utilized as the pruning method. As observed, integrating VID (Ahn et al., 2019) further improves training on the pruned dataset.

## D. Impact of KD temperature

In Section 4.4 we have made the observation that for low pruning fractions, employing KD using smaller teachers results in higher student accuracy. To demonstrate the consistency of this observation across different KD temperatures, in Figure 11 we present the impact of the KD temperature on the student's accuracy when utilizing teachers with different capacities, and across various pruning fractions. The experiment was conducted on CIFAR-100 with random pruning using a ResNet-20 student. As can be observed, the benefit of smaller teachers in high pruning regimes (lower $f$ values) is evident over a wide range of temperature values.

## E. Comparing different KD approaches

So far, we have utilized solely vanilla KD during training. Next we explore integrating additional KD approaches to the loss. In particular, we add an additional KD loss term $\mathcal{L}_R$ as follows: $\mathcal{L}(\theta) = \mathcal{L}_{\mathrm{cls}}(\theta) + \alpha \mathcal{L}_{\mathrm{KD}}(\theta) + \beta \mathcal{L}_R(\theta)$, where $\beta$ is a hyper-parameter. In this experiments, we simply set $\alpha$ and $\beta$ to 1. In Table 1 we compare the performance of different KD methods on CIFAR-100 under low and average compression regimes. For a fair comparison, for the case of employing only the vanilla KD, we set $\alpha = 2$, and $\beta = 0$. As can be observed, integrating the Variational Information Distillation (VID) loss (Ahn et al., 2019) improves results considerably for the tested cases. These results suggest that further improvement can be achieved by incorporating additional approaches to extract knowledge from the teacher.

## F. Impact of pruning levels

In this section, we present results comparing 'easy,' 'moderate' and 'hard' pruning levels when integrating knowledge distillation (KD) into the loss function. Figure 12 illustrates the accuracy achieved on CIFAR-100 across the three pruning

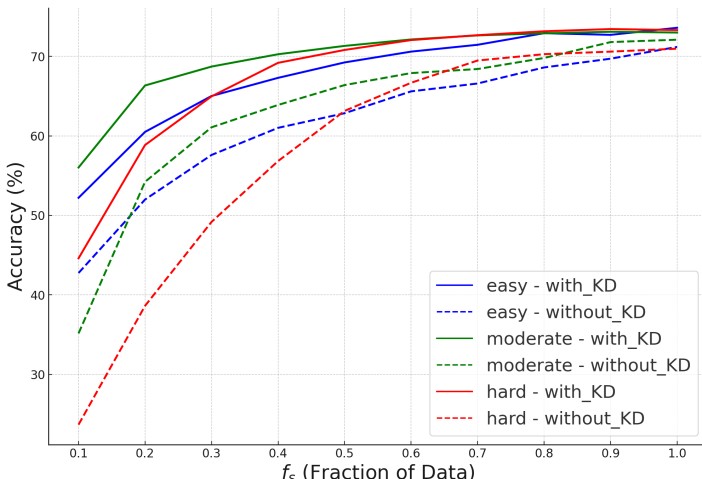

*Figure 12.* **Pruning levels (easy, moderate, and hard pruning)**. In easy (hard) pruning, we select the $f$-percentile of lowest (highest) scores. Moderate pruning refers to selecting the middle $f$-percentile. This figure reveals multiple insights: (1) easy and moderate pruning produce higher results compared to hard pruning for low pruning fractions (both with and without KD); (2) using KD, moderate pruning leads to top performance compared to 'easy' and 'hard' pruning levels; and (3) using KD, the variance between pruning levels is reduced. These results were obtained on CIFAR-100.

levels. Specifically, we employed the *forgetting* approach to compute a score for each training sample. For 'easy' pruning, we selected the $f$-percentile of samples with the lowest scores, while for 'hard' pruning, we selected the $f$-percentile of samples with the highest scores. 'Moderate' pruning involved selecting samples within the middle $f$-percentile. The results highlight several key insights: (1) both 'easy' and 'moderate' pruning outperform 'hard' pruning in terms of accuracy (with and without KD) in low pruning fractions; (2) incorporating KD, 'moderate' pruning achieves the highest performance compared to 'easy' and 'hard' pruning; and (3) KD reduces the variance in performance across the different pruning levels.

## G. Theoretical Motivation

**Lemma G.1.** *Given a data matrix $\mathbf{X} \in \mathbb{R}^{d \times N}$ and its sub-matrix $\mathbf{X}_f \in \mathbb{R}^{d \times N_f}$, while $d \leq N_f \leq N$,*

$$\sigma_k(\mathbf{X}) \geq \sigma_k(\mathbf{X}_f), k = 1, \ldots, d,$$

*where $\sigma_k(\mathbf{X})$ is the $k$'s largest singular value of $\mathbf{X}$.*

*Proof.* Let $\mathbf{Z}$ denote the remaining sub-matrix after excluding the $\mathbf{X}_f$ columns from $\mathbf{X}$, i.e., , $\mathbf{X} = [\mathbf{X}_f | \mathbf{Z}]$. Thus,

$$\mathbf{X}\mathbf{X}^T = \mathbf{X}_f\mathbf{X}_f^T + \mathbf{Z}\mathbf{Z}^T.$$

All three matrices are positive semidefinite and therefore based on Weyl's inequality (Horn & Johnson, 2012)(Theorem 4.3.1), $\lambda_k(\mathbf{X}\mathbf{X}^T) \geq \lambda_k(\mathbf{X}_f\mathbf{X}_f^T)$, where $\lambda_k(\mathbf{A})$ is the $k$'s largest eigenvalue of $\mathbf{A}$. This also implies that $\sigma_k(\mathbf{X}) \geq \sigma_k(\mathbf{X}_f)$ for $k = 1, \ldots, d$. □

**Theorem G.2.** *Let $\mathbf{X} \in \mathbb{R}^{d \times N}$ and $\mathbf{y} \in \mathbb{R}^N$ denote the observations matrix and ground-truth label vector, respectively. Let $\hat{\boldsymbol{\theta}}_s(\alpha, f, f_t)$ denote the student model obtained using Eq. 4 using pruning factor $f < f_t$ and distilled from the teacher model $\hat{\boldsymbol{\theta}}(f_t)$ using KD weight $\alpha$. Then, the following holds,*

$$||\mathbb{E}_\eta[\hat{\boldsymbol{\epsilon}}_s(\alpha, f, f_t)]||^2 \leq ||\mathbb{E}_\eta[\hat{\boldsymbol{\epsilon}}_s(\alpha, f, f)]||^2.$$

*Proof.* Similarly to (Das & Sanghavi, 2023) we base our proof on the Singular Value Decomposition (SVD) of both the pruned and the full data matrices used to train the student and the teacher, respectively. Thus, $\mathbf{X}_{f_t} = \mathbf{U}'\mathbf{\Sigma}'\mathbf{V}'^T$ and

$\mathbf{X}_f = \mathbf{U}\mathbf{\Sigma}\mathbf{V}^T$. We also assume that $N > N_f \geq d$, which is a practical assumption in machine learning and therefore the rank of both the full and the pruned data matrices is $d$. Thus the estimator SVD has the following form in terms of the SVD of the full and pruned data matrices,

$$\hat{\boldsymbol{\theta}}_s(\alpha, f, f_t) = (\mathbf{X}_f\mathbf{X}_f^T + \lambda\mathbf{I}_d)^{-1}\mathbf{X}_f\big((1-\alpha)\mathbf{y}_f + \alpha\mathbf{X}_f^T\hat{\boldsymbol{\theta}}(f_t)\big)$$

$$= \mathbf{U}\left(\mathbf{\Sigma}^2 + \lambda\mathbf{I}_d\right)^{-1}\mathbf{\Sigma}\left((1-\alpha)(\mathbf{\Sigma}\mathbf{U}^T\boldsymbol{\theta}^* + \mathbf{V}^T\boldsymbol{\eta}_f) + \alpha\mathbf{\Sigma}\mathbf{U}^T\hat{\boldsymbol{\theta}}(f_t)\right)$$

$$= \mathbf{U}\left(\mathbf{\Sigma}^2 + \lambda\mathbf{I}_d\right)^{-1}\mathbf{\Sigma}\Big((1-\alpha)(\mathbf{\Sigma}\mathbf{U}^T\boldsymbol{\theta}^* + \mathbf{V}^T\boldsymbol{\eta}_f) +$$

$$+ \alpha\mathbf{\Sigma}\mathbf{U}^T\mathbf{U}'\left(\mathbf{\Sigma}'^2 + \lambda\mathbf{I}_d\right)^{-1}\mathbf{\Sigma}'\left(\mathbf{\Sigma}'\mathbf{U}'^T\boldsymbol{\theta}^* + \mathbf{V}'^T\boldsymbol{\eta}_{f_t}\right)\Big)$$

$$= \sum_{i=1}^{d}\frac{\sigma_i^2}{\sigma_i^2 + \lambda}\left((1-\alpha)\langle\boldsymbol{\theta}^*, \mathbf{u}_i\rangle + \alpha\sum_{j=1}^{d}\frac{\sigma_j'^2}{\sigma_j'^2 + \lambda}\langle\boldsymbol{\theta}^*, \mathbf{u}_j'\rangle\langle\mathbf{u}_j', \mathbf{u}_i\rangle\right)\mathbf{u}_i +$$

$$+ \sum_{i=1}^{d}\frac{\sigma_i}{\sigma_i^2 + \lambda}\left((1-\alpha)\langle\boldsymbol{\eta}_f, \mathbf{v}_i\rangle + \alpha\sigma_i\sum_{j=1}^{d}\frac{\sigma_j'}{\sigma_j'^2 + \lambda}\langle\boldsymbol{\eta}_{f_t}, \mathbf{v}_j'\rangle\langle\mathbf{u}_j', \mathbf{u}_i\rangle\right)\mathbf{u}_i.$$

$$= \sum_{i=1}^{d}\frac{\sigma_i^2}{\sigma_i^2 + \lambda}\left((1-\alpha)\langle\sum_{j=1}^{d}\langle\boldsymbol{\theta}^*, \mathbf{u}_j'\rangle\mathbf{u}_j', \mathbf{u}_i\rangle + \alpha\sum_{j=1}^{d}\frac{\sigma_j'^2}{\sigma_j'^2 + \lambda}\langle\boldsymbol{\theta}^*, \mathbf{u}_j'\rangle\langle\mathbf{u}_j', \mathbf{u}_i\rangle\right)\mathbf{u}_i$$

$$+ \sum_{i=1}^{d}\frac{\sigma_i}{\sigma_i^2 + \lambda}\left((1-\alpha)\langle\boldsymbol{\eta}_f, \mathbf{v}_i\rangle + \alpha\sigma_i\sum_{j=1}^{d}\frac{\sigma_j'}{\sigma_j'^2 + \lambda}\langle\boldsymbol{\eta}_{f_t}, \mathbf{v}_j'\rangle\langle\mathbf{u}_j', \mathbf{u}_i\rangle\right)\mathbf{u}_i.$$

$$= \sum_{i=1}^{d}\frac{\sigma_i^2}{\sigma_i^2 + \lambda}\left((1-\alpha)\sum_{j=1}^{d}\langle\boldsymbol{\theta}^*, \mathbf{u}_j'\rangle\langle\mathbf{u}_j', \mathbf{u}_i\rangle + \alpha\sum_{j=1}^{d}\frac{\sigma_j'^2}{\sigma_j'^2 + \lambda}\langle\boldsymbol{\theta}^*, \mathbf{u}_j'\rangle\langle\mathbf{u}_j', \mathbf{u}_i\rangle\right)\mathbf{u}_i$$

$$+ \sum_{i=1}^{d}\frac{\sigma_i}{\sigma_i^2 + \lambda}\left((1-\alpha)\langle\boldsymbol{\eta}_f, \mathbf{v}_i\rangle + \alpha\sigma_i\sum_{j=1}^{d}\frac{\sigma_j'}{\sigma_j'^2 + \lambda}\langle\boldsymbol{\eta}_{f_t}, \mathbf{v}_j'\rangle\langle\mathbf{u}_j', \mathbf{u}_i\rangle\right)\mathbf{u}_i.$$

$$= \sum_{i=1}^{d}\sum_{j=1}^{d}\frac{\sigma_i^2}{\sigma_i^2 + \lambda}\langle\boldsymbol{\theta}^*, \mathbf{u}_j'\rangle\langle\mathbf{u}_j', \mathbf{u}_i\rangle\left(1 - \alpha\frac{\lambda}{\sigma_j'^2 + \lambda}\right)\mathbf{u}_i +$$

$$+ \sum_{i=1}^{d}\frac{\sigma_i}{\sigma_i^2 + \lambda}\left((1-\alpha)\langle\boldsymbol{\eta}_f, \mathbf{v}_i\rangle + \alpha\sigma_i\sum_{j=1}^{d}\frac{\sigma_j'}{\sigma_j'^2 + \lambda}\langle\boldsymbol{\eta}_{f_t}, \mathbf{v}_j'\rangle\langle\mathbf{u}_j', \mathbf{u}_i\rangle\right)\mathbf{u}_i.$$

The estimation error is therefore,

$$\hat{\boldsymbol{\epsilon}}_s(\alpha, f, f_t) = \hat{\boldsymbol{\theta}}_s(\alpha, f, f_t) - \boldsymbol{\theta}^* = \hat{\boldsymbol{\theta}}_s(\alpha, f, f_t) - \sum_{i=1}^{d}\langle\boldsymbol{\theta}^*, \mathbf{u}_i\rangle\mathbf{u}_i$$

$$= \hat{\boldsymbol{\theta}}_s(\alpha, f, f_t) - \sum_{i=1}^{d}\langle\sum_{j=1}^{d}\langle\boldsymbol{\theta}^*, \mathbf{u}_j'\rangle\mathbf{u}_j', \mathbf{u}_i\rangle\mathbf{u}_i$$

$$= \sum_{i=1}^{d}\sum_{j=1}^{d}\frac{\sigma_i^2}{\sigma_i^2 + \lambda}\langle\boldsymbol{\theta}^*, \mathbf{u}_j'\rangle\langle\mathbf{u}_j', \mathbf{u}_i\rangle\left(1 - \alpha\frac{\lambda}{\sigma_j'^2 + \lambda}\right)\mathbf{u}_i - \sum_{i=1}^{d}\sum_{j=1}^{d}\langle\boldsymbol{\theta}^*, \mathbf{u}_j'\rangle\langle\mathbf{u}_j', \mathbf{u}_i\rangle\mathbf{u}_i$$

$$+ \sum_{i=1}^{d}\frac{\sigma_i}{\sigma_i^2 + \lambda}\left((1-\alpha)\langle\boldsymbol{\eta}_f, \mathbf{v}_i\rangle + \alpha\sigma_i\sum_{j=1}^{d}\frac{\sigma_j'}{\sigma_j'^2 + \lambda}\langle\boldsymbol{\eta}_{f_t}, \mathbf{v}_j'\rangle\langle\mathbf{u}_j', \mathbf{u}_i\rangle\right)\mathbf{u}_i.$$

$$= \sum_{i=1}^{d}\sum_{j=1}^{d}\langle\boldsymbol{\theta}^*, \mathbf{u}_j'\rangle\langle\mathbf{u}_j', \mathbf{u}_i\rangle\left(\frac{\sigma_i^2}{\sigma_i^2 + \lambda}\left(1 - \alpha\frac{\lambda}{\sigma_j'^2 + \lambda}\right) - 1\right)\mathbf{u}_i$$

$$+ \sum_{i=1}^{d} \frac{\sigma_i}{\sigma_i^2 + \lambda} \left( (1-\alpha)\langle \boldsymbol{\eta}_f, \mathbf{v}_i \rangle + \alpha \sigma_i \sum_{j=1}^{d} \frac{\sigma_j'}{\sigma_j'^2 + \lambda} \langle \boldsymbol{\eta}_{f_t}, \mathbf{v}_j' \rangle \langle \mathbf{u}_j', \mathbf{u}_i \rangle \right) \mathbf{u}_i$$

$$= -\sum_{i=1}^{d} \sum_{j=1}^{d} \langle \boldsymbol{\theta}^*, \mathbf{u}_j' \rangle \langle \mathbf{u}_j', \mathbf{u}_i \rangle \frac{\lambda}{\sigma_i^2 + \lambda} \left( 1 + \alpha \frac{\sigma_i^2}{\sigma_j'^2 + \lambda} \right) \mathbf{u}_i$$

$$+ \sum_{i=1}^{d} \frac{\sigma_i}{\sigma_i^2 + \lambda} \left( (1-\alpha)\langle \boldsymbol{\eta}_f, \mathbf{v}_i \rangle + \alpha \sigma_i \sum_{j=1}^{d} \frac{\sigma_j'}{\sigma_j'^2 + \lambda} \langle \boldsymbol{\eta}_{f_t}, \mathbf{v}_j' \rangle \langle \mathbf{u}_j', \mathbf{u}_i \rangle \right) \mathbf{u}_i.$$

The expectation of the bias term over the noise parameter $\eta$ which is uncorrelated and independent of $\mathbf{X}$ is,

$$\mathbb{E}_\eta[\hat{\boldsymbol{\epsilon}}_s(\alpha, f, f_t)] = -\sum_{i=1}^{d} \sum_{j=1}^{d} \langle \boldsymbol{\theta}^*, \mathbf{u}_j' \rangle \langle \mathbf{u}_j', \mathbf{u}_i \rangle \frac{\lambda}{\sigma_i^2 + \lambda} \left( 1 + \alpha \frac{\sigma_i^2}{\sigma_j'^2 + \lambda} \right) \mathbf{u}_i.$$

Therefore the bias error term of the estimation process is,

$$||\mathbb{E}_\eta[\hat{\boldsymbol{\epsilon}}_s(\alpha, f, f_t)]||^2 = \sum_{i=1}^{d} \left( \frac{\lambda}{\sigma_i^2 + \lambda} \right)^2 \left( \sum_{j=1}^{d} \langle \boldsymbol{\theta}^*, \mathbf{u}_j' \rangle \langle \mathbf{u}_j', \mathbf{u}_i \rangle \left( 1 + \alpha \frac{\sigma_i^2}{\sigma_j'^2 + \lambda} \right) \right)^2.$$

Note that given that the student and the teacher are trained using the same dataset $\mathbf{X}_f$, i.e., , $\sigma_i = \sigma_i'$ and $\mathbf{u}_i = \mathbf{u}_i'$ for $i = 1, \ldots, d$, the bias error term reduces to what is reported in (Das & Sanghavi, 2023) (Eq. 24):

$$||\mathbb{E}_\eta[\hat{\boldsymbol{\epsilon}}_s(\alpha, f, f)]||^2 = \sum_{i=1}^{d} \left( \frac{\lambda}{\sigma_i^2 + \lambda} \right)^2 \left( \sum_{j=1}^{d} \langle \boldsymbol{\theta}^*, \mathbf{u}_j \rangle \langle \mathbf{u}_j, \mathbf{u}_i \rangle \left( 1 + \alpha \frac{\sigma_i^2}{\sigma_j^2 + \lambda} \right) \right)^2$$

$$= \sum_{i=1}^{d} \langle \boldsymbol{\theta}^*, \mathbf{u}_i \rangle^2 \left( \frac{\lambda}{\sigma_i^2 + \lambda} \right)^2 \left( 1 + \alpha \frac{\sigma_i^2}{\sigma_j^2 + \lambda} \right)^2.$$

Now, let us consider the impact of a minimal augmentation of the dataset used to train the teacher w.r.t. that used to train the student. In other words, we assume that a single data sample is added, i.e., $f_t = f + \frac{1}{N}$, where $N$ is the total number of available samples. Given that adding a single sample to a significantly larger set of $N_f$ samples is not sufficient to change its distribution, we can assume that $\mathbf{u}_i' \approx \mathbf{u}_i$ for $i = 1, \ldots, d$. Thus, the derivative of the error bias term with respect to $\sigma_k'$ is,

$$\frac{\partial ||\mathbb{E}_\eta[\hat{\boldsymbol{\epsilon}}_s(\alpha, f, f + \frac{1}{N})]||^2}{\partial \sigma_k'} = 2 \sum_{i=1}^{d} \left( \frac{\lambda}{\sigma_i^2 + \lambda} \right)^2 \sum_{j=1}^{d} \langle \boldsymbol{\theta}^*, \mathbf{u}_j' \rangle \langle \mathbf{u}_j', \mathbf{u}_i \rangle \cdot$$

$$\cdot \left( 1 + \alpha \frac{\sigma_i^2}{\sigma_j'^2 + \lambda} \right) \left( -\alpha \frac{2\sigma_k' \sigma_i^2}{(\sigma_k'^2 + \lambda)^2} \right) \langle \boldsymbol{\theta}^*, \mathbf{u}_k' \rangle \langle \mathbf{u}_k', \mathbf{u}_i \rangle$$

$$\approx -4\alpha \left( \frac{\lambda}{\sigma_k^2 + \lambda} \right)^2 \left( 1 + \alpha \frac{\sigma_k^2}{\sigma_j'^2 + \lambda} \right) \frac{\sigma_k' \sigma_k^2}{(\sigma_k'^2 + \lambda)^2} \langle \boldsymbol{\theta}^*, \mathbf{u}_k' \rangle^2$$

$$\leq 0.$$

According to Lemma G.1, $\sigma_k(\mathbf{X}_{f+\frac{1}{N}}) \geq \sigma_k(\mathbf{X}_f), \forall k = 1, \ldots, d$. Since we have shown that $\frac{\partial ||\mathbb{E}_\eta[\hat{\boldsymbol{\epsilon}}_s(\alpha, f, f+\frac{1}{N})]||^2}{\partial \sigma_k(\mathbf{X}_{f+\frac{1}{N}})} \leq 0$, i.e., , the derivative of the error bias term w.r.t a singular value $\sigma_k'$ of the teacher data matrix $\mathbf{X}_{f_t}$ is non-positive, and the pruned data matrix used to train the student necessarily has smaller corresponding singular values, it necessarily implies that $||\mathbb{E}_\eta[\hat{\boldsymbol{\epsilon}}_s(\alpha, f, f+\frac{1}{N})]||^2 \leq ||\mathbb{E}_\eta[\hat{\boldsymbol{\epsilon}}_s(\alpha, f, f)]||^2$. Applying the same logic iteratively over the process of adding more and more data samples, implies that $||\mathbb{E}_\eta[\hat{\boldsymbol{\epsilon}}_s(\alpha, f, f_t)]||^2 \leq ||\mathbb{E}_\eta[\hat{\boldsymbol{\epsilon}}_s(\alpha, f, f)]||^2$ for any $f_t > f$. $\qquad \square$

