# OpenReview forum: "Distilling the Knowledge in Data Pruning"
_ICML.cc/2025/Conference — ICML 2025 poster_

### Official Review · Reviewer_xePE · 2025-03-03

**Overall Recommendation:** 4

**Summary:**

Dataset pruning is the task of reducing the number of samples within a dataset without impairing accuracy. This article combines dataset pruning with methods inspired by the knowledge distillation (KD) literature by augmenting the training process with soft labels from a teacher network that was trained on the full dataset. As a result, the approach can be used to improve existing pruning methods (higher accuracy with the same number of samples / matched accuracy with fewer samples).

## Update After Rebuttal:
The authors were responsive and addressed my concerns, thus I'm in favor of acceptance.

**Claims And Evidence:**

To my knowledge, the first paper that combines KD with dataset pruning. Most of the empirical results are done on small-scale datasets (CIFAR, SVHN) which is a bit of a drawback. At the same time, pruning experiments are expensive, and not all academic labs have the resources to run dozens / hundreds of ImageNet training runs, thus I don't think this should be held against the paper.

The empirical results are strong; KD leads to strongly improved accuracies and can be combined with existing pruning methods, which is nice. KD of course comes with the limitation that one needs to train a full model on the entire dataset first; this is definitely a limiting factor for some scenarios but nonetheless, if one would like to run lots of ablations / variants one could train the full model once, and then run the ablations on a smaller subset of the dataset, thus there are certain cases where KD can be helpful and still save compute.

Experimentally, **my biggest concern is that results are compared against the baseline of "100% data, no KD". If KD is proposed as part of the training procedure, shouldn't a stronger baseline be the case of "100% data, KD"?** If so, which methods / pruning metrics still maintain accuracy w.r.t. this stronger baseline?

**Essential References Not Discussed:**

N/A.

**Experimental Designs Or Analyses:**

I checked the soundness and validity of the experimental design. Overall it is sound; the two downsides to the employed approach are that (1.) a teacher [trained on the full dataset] is needed; (2.) the approach introduces an additional hyperparameter $\alpha$ controlling the loss weight (cf. Figure 6). However, none of those two downsides invalidate the approach.

**Methods And Evaluation Criteria:**

The benchmark datasets and comparison metrics are standard in the literature (accuracy with varying pruning fractions). The core idea - combining KD with dataset pruning - is compellingly simple and straightforward.

**Other Comments Or Suggestions:**

- Would recommend a pass checking for \cite{}, \citet{}, \citep{} differences - e.g. when talking about a specific paper by Authors (YYYY) this should not be cited as (Authors, YYYY) in the text.
- Nit: Figure 5 has a different plotting style compared to other figures (e.g. grid). Furthermore, I would suggest to use a sequential color palette for Figure 5 due to the nature of the sequential data.

**Other Strengths And Weaknesses:**

The paper is well written overall, accessible, and clear. Figure 1 is a great overview; I found in particular subfigure 1c intriguing since it's a very systematic exploration, with a clear (and at first sight, counter-intuitive) result.

**Questions For Authors:**

Major:
- [copied from above] Experimentally, my biggest concern is that results are compared against the baseline of "100% data, no KD". If KD is proposed as part of the training procedure, shouldn't a stronger baseline be the case of "100% data, KD"? If so, which methods / pruning metrics still maintain accuracy w.r.t. this stronger baseline?

Minor:
- Given that the authors mentioned that they used several metrics from https://github.com/rgeirhos/dataset-pruning-metrics, why not the SSL prototypes metric (which is one of the best-performing ones)?

**Relation To Broader Scientific Literature:**

To my knowledge, this is the first paper that combines KD with dataset pruning. The focus is on pruning, while the approach is based on KD. I appreciate the transfer / bridging of those two related subfields.

**Theoretical Claims:**

I did not carefully assess the theory since dataset pruning is very much an empirical field; while a theoretical motivation is great the ultimate test that decides whether a dataset pruning metric will be useful are experimental results.

---

> ### Author Rebuttal · Authors · 2025-03-31
>
> We sincerely thank **Reviewer xePE** for the thoughtful reading, positive feedback and appreciation for the novelty and simplicity of our proposed approach. Below, we kindly address each of the reviewer's questions and concerns:
>
> **1. Comparison with a Stronger Baseline (100% Data, KD)**
>
> We thank the reviewer for this question, and kindly note that the baseline they refer to (100% data, KD) is included already in the paper. As part of our CIFAR-100 and ImageNet experiments (figures 3a, 4a, 4b), this baseline is represented as the joint plot point for all experiments which utilize $f=1$ with KD (and regardless of the pruning method). We emphasize that this plot point is joint for all KD experiments regardless of the pruning method, since when the entire dataset is utilized ($f=1$) no pruning is being conducted, and hence the selected pruning approach has no effect. Following the reviewer’s suggestion, we will include the exact accuracy numbers in the main paper.
>
> In addition, as can be observed in the aforementioned figures, performance can be mostly preserved w.r.t to this strong baseline across several pruning factors when using several pruning methods with KD. For example, on CIFAR-100, performance is mostly preserved when combining most pruning methods with KD and retaining only 60% of the data.
> Finally, we note that this stronger baseline is not shown for the SVHN and CIFAR-10 experiments (figures 3b, 3c), since, as can be observed in the aforementioned figures, model performance gets saturated very quickly on these datasets. Hence, performance on higher pruning factors was omitted for these datasets for visualization purposes.
>
> **2. Comparison to the SSL Prototypes Method [1]**
>
> While we haven’t compared our approach to this specific method, we have recently compared it with other (more recent) approaches, namely: **Moderate-DS [2]**, **D2 [3]**, and **DUAL [4]**. The comparison results can be viewed in this figure:
>
> [experiments_recent_pruning_methods.png](https://ibb.co/xKB0F9hq)
>
> As depicted in the figure, all pruning methods (including the recent ones) benefit from the incorporation of KD in the training process. In addition, it can be seen that on low pruning fractions, random pruning + KD maintains its edge over all other methods, except for D2 + KD which achieves a similar performance. However, contrary to D2 which requires careful tuning of several hyperparameters (k,$\gamma_{r}$​,β) for each pruning fraction and for each dataset, our method (simple random pruning + KD) has no such requirements and is hence more practical and easier to adopt for real-world use cases.
>
> **3. Citation Formatting Issues**
>
> We thank the reviewer for bringing these issues to our attention. These issues will be corrected in the final version of the paper.
>
> **4. Figure 5 Style**
>
> Following the reviewer’s suggestion, we have revised Figure 5 by using a sequential color palette and adjusting the figure style to match the other figures in the paper. The revised figure can be observed here:
>
> [updated_figure_5_accuracy_vs_teacher_data_fraction.png](https://ibb.co/cXCVqZnK)
>
> We thank the reviewer for this comment.
>
> $~$
>
> **[1]** Beyond neural scaling laws: beating power law scaling via data pruning, NeurIPS, 2022.
>
> **[2]** Moderate Coreset: A Universal Method of Data Selection for Real-world Data-efficient Deep Learning, ICLR, 2023.
>
> **[3]** D2 Pruning: Message Passing for Balancing Diversity & Difficulty in Data Pruning, ICLR, 2024.
>
> **[4]** Lightweight Dataset Pruning without Full Training via Example Difficulty and Prediction Uncertainty, ArXiv, 2025.

---

> > ### Comment · Reviewer_xePE · 2025-04-03
> >
> > I would like to thank the authors for taking the time to respond.
> >
> > I appreciate the broadening to more comparison methods (2) and the smaller changes (3) and (4).
> >
> > Regarding the the baseline (1), sorry if my initial review wasn't clear in this regard. I understand that the datapoint is shown in Figures 3a,4a,4b already; my suggestion/concern was that this datapoint should be the reference point against which performance is compared. Currently, the horizontal dashed line in those plots (which serves as the visual baseline for comparison) corresponds to 100% data, teacher accuracy (*without* KD) which I find less than ideal; I'm suggesting to change this dashed line to the more appropriate reference of 100% data *with* KD. Correspondingly, this also affects the description of results in the paper.
> >
> > As a separate comment, others (in a discussion not visible to the authors) have pointed out that there is some prior work connecting pruning to distillation, such as:
> >
> > Moser, Brian B., et al. "Distill the Best, Ignore the Rest: Improving Dataset Distillation with Loss-Value-Based Pruning." arXiv preprint arXiv:2411.12115 (2024).
> >
> > Sundar, Anirudh S., et al. "Prune then distill: Dataset distillation with importance sampling." ICASSP 2023-2023 IEEE International Conference on Acoustics, Speech and Signal Processing (ICASSP). IEEE, 2023
> >
> > I should have done a more thorough literature search for my initial review which was based on the assumption that combining pruning and distillation techniques is a novel combination; given that there is some prior work in this space I'm still in favor of acceptance (in light of interesting and strong experimental results) albeit a bit less enthusiastically than before. I would encourage the authors to broaden their discussion of related work (including e.g. works that combine dataset distillation with pruning methods).

---

> > > ### Author Response · Authors · 2025-04-03
> > >
> > > We thank the reviewer for their thoughtful response and truly appreciate the time and effort invested in evaluating our submission.
> > >
> > > &nbsp;
> > >
> > > **1.**
> > > We now understand your request and thank you for the clarification. We will certainly include the horizontal line representing the case of 100% of the data with KD in both the figures and the corresponding text. We agree with the reviewer’s observation that this representation would serve as a more appropriate baseline. In our current figures, the dashed horizontal line was set to reference the teacher accuracy. However, we acknowledge that using the case of 100% data with KD as the baseline would be more informative and relevant, as suggested by the reviewer. We will make the necessary adjustments accordingly.
> > >
> > > &nbsp;
> > >
> > > **2.**
> > > Please note that the mentioned papers [1], [2] focus on dataset distillation, which, while related,  fundamentally differs from data pruning or knowledge distillation. Dataset distillation aims to generate a compact set of synthetic samples (e.g., images) using optimization techniques such as trajectory matching or distribution matching. In contrast, our work takes a different approach by integrating knowledge distillation into the classification loss when training on a pruned dataset, utilizing the soft predictions provided by a teacher model.
> > >
> > > While we have already discussed the connections and differences between dataset distillation and dataset pruning in the related work section, we appreciate the reviewer’s suggestion. We will enhance our discussion to include additional works that address both dataset distillation and pruning, providing a more comprehensive context for our approach.
> > >
> > > In addition, we believe that our work’s novelty lies in utilizing knowledge distillation within data pruning while making several valuable and intriguing observations, and providing theoretical motivation. To the best of our knowledge, these insights have not been previously explored. Specifically, we demonstrate that in the presence of KD, model accuracy remains robust regardless of the data pruning method employed. This finding has practical implications, as it suggests that simple random pruning, when combined with KD, is a viable alternative to more sophisticated pruning methods. Interestingly, we also observe that increasing the teacher model size can lead to a decrease in accuracy when dealing with small pruning fractions.
> > >
> > >
> > > &nbsp;
> > >
> > > We hope this response clarifies our approach and addresses the reviewer’s concerns. Once again, we thank the reviewer for the constructive feedback, which will undoubtedly help improve the quality of our work.
> > >
> > > &nbsp;
> > >
> > >
> > > **[1]** Moser, Brian B., et al. “Distill the Best, Ignore the Rest: Improving Dataset Distillation with Loss-Value-Based Pruning.” arXiv preprint arXiv:2411.12115 (2024).
> > >
> > > **[2]** Sundar, Anirudh S., et al. “Prune then distill: Dataset distillation with importance sampling.” ICASSP 2023-2023 IEEE International Conference on Acoustics, Speech and Signal Processing (ICASSP). IEEE, 2023 (edited)

---

### Official Review · Reviewer_J77D · 2025-03-13

**Overall Recommendation:** 3

**Summary:**

The key contribution of this paper is to show that using soft predictions from teachers trained on complete data combined with prune data improves the accuracy of students trained on pruned data. The authors conduct a series of experiments with ResNet variations on small datasets, including CIFAR-10, CIFAR-100, and SVHN, to demonstrate this. The results clearly support the authors' claims.

The paper is easy to follow. The experiments support the claim that using knowledge distillation (KD) improves the performance of students trained on pruned datasets. The authors also provide theoretical motivation for why KD of teacher trained on complete data is helpful.

**Claims And Evidence:**

The claim in the paper was supported with experiments, however I have one main concern as below.

**Essential References Not Discussed:**

n/a

**Experimental Designs Or Analyses:**

See above.

**Methods And Evaluation Criteria:**

While the paper presents some observations that I have not seen before, my main concern is the practicality of the setup. If the teacher has already been trained on complete data, what is the value of using pruned data to train the student, given that most of the knowledge can come from the teacher? It seems to me that this approach is only useful if the pruned data is not a subset of the complete dataset or if the authors can demonstrate the value of using pruned data in this setup. Could the authors provide either comparison as follows?

1. Conduct similar experiments to show improvements when the pruned data is not a subset of the complete dataset used to train the teacher.

2. Or provide results where the student is trained solely using the teacher’s soft predictions, without pruned data (I assume this corresponds to f = 0?), as a baseline. This would help demonstrate the value of using pruned data to train the student.

The temperature parameter is quite important when training with knowledge distillation (KD). Therefore, I suggest that the authors include ablation studies on temperatures in these experiments to ensure the best temperature is selected for fair comparisons.

Because of this, I tend to rate it as weak reject but happy to increase the rating if my concern is addressed.

**Other Comments Or Suggestions:**

see above

**Other Strengths And Weaknesses:**

n/a

**Questions For Authors:**

see above

**Relation To Broader Scientific Literature:**

The paper might have the small impacts

**Theoretical Claims:**

The theoretical proof sounds and seems supports the the claim, but I have not check the correctness of the proofs.

---

> ### Author Rebuttal · Authors · 2025-03-31
>
> We sincerely thank the reviewer for their valuable feedback and for highlighting the strengths of our work. We appreciate the constructive suggestions and are committed to addressing the concerns raised.
>
> &nbsp;
>
> **Experimental Results with Disjoint Datasets**
>
> Following the reviewer's suggestion, we conducted experiments to evaluate the case where the pruned data is not a subset of the complete dataset used to train the teacher.
>
>
> Let $\mathcal{P}$  be a pruned dataset sampled from $\mathcal{D}$ to train the student model, and let $\mathcal{S}$ be the training data for the teacher. In the following experiments, $\mathcal{D}$ and $\mathcal{S}$ are disjoint i.e.,  $\mathcal{P} \cap \mathcal{S} = \emptyset$.
>
> For the empirical study, we used 70% of the training data to train the teacher and the remaining 30% to train the student with different pruning ratios. Specifically, we compared the performance with and without knowledge distillation (KD) for CIFAR-100 and SVHN datasets.
>
> The experimental results can be viewed in the following link:
>
> [Experimental_Results_with_Disjoint_datasets.png](https://ibb.co/dwP4fDyf)
>
> Notably, combining knowledge distillation (KD) with data pruning yields significant performance gains, even when the student is trained on a pruned dataset that differs from the teacher's training data. For instance, in CIFAR-100 with random pruning at $f=50$%, we observe a 14.5-point accuracy improvement when the teacher model was trained on a different subset.
>
> &nbsp;
>
> We are happy to include these experimental results in the main paper. We believe that these findings further support our proposed approach, particularly in the context discussed at the end of Section 3.1, namely use cases where the full dataset is no longer accessible (e.g. due to privacy concerns).
>
> &nbsp;
>
> **Ablation Studies on Temperature**
>
> Following the reviewer's suggestion to include ablation studies on temperature, we kindly refer to Figure 10 in the appendix, where we present accuracy results for various temperature values across different pruning fractions and architectures.
>
> &nbsp;
>
>
> We hope our response addresses your concerns, and we would greatly appreciate it if you could consider raising your rating of our work.

---

> > ### Comment · Reviewer_J77D · 2025-04-05
> >
> > Thank you to the authors for the rebuttal. It has addressed my concerns in the review, so I am increasing my rating to Weak Accept from my initial rating of Weak Reject.

---

> > > ### Author Response · Authors · 2025-04-07
> > >
> > > We thank the reviewer for their willingness to raise the score.

---

### Official Review · Reviewer_RfVw · 2025-03-14

**Overall Recommendation:** 3

**Summary:**

The paper explores the use of knowledge distillation (KD) for enhancing training on pruned datasets, demonstrating that simple random pruning with KD can achieve superior accuracy compared to recent data pruning methods. The work also reveals that, when using teachers with smaller capacities, the student can be more beneficial in low pruning fractions. The study provides theoretical motivation and empirical evidence, showing that KD helps mitigate the impact of label noise and improve accuracy.

**Claims And Evidence:**

The main claim of this paper is that a simple random pruning method, when combined with knowledge distillation (KD), outperforms all standard data pruning algorithms that rely on hard labels, and this claim is supported by clear and convincing evidence.

**Essential References Not Discussed:**

While the paper introduces a simple yet effective approach in data pruning literature, several of recent works are not covered, including:
- Moderate Coreset: A Universal Method of Data Selection for Real-world Data-efficient Deep Learning, ICLR, 2022
- Robust Data Pruning under Label Noise via Maximizing Re-labeling Accuracy, NeurIPS, 2023
- D^2  Pruning: Message Passing for Balancing Diversity & Difficulty in Data Pruning, ICLR, 2024
- Lightweight Dataset Pruning without Full Training via Example Difficulty and Prediction Uncertainty, ArXiv, 2025

**Experimental Designs Or Analyses:**

The experiment setup is extensive and well-constructed.

**Methods And Evaluation Criteria:**

The proposed method is technically sound, and the extensive experiments on multiple datasets including ImageNet demonstrate its superiority.

**Other Comments Or Suggestions:**

Can the idea of using knowledge distillation for data pruning also be applied to data pruning for vision-language models (VLM) [1] or large language models (LLM) [2]?

[1] Too Large; Data Reduction for Vision-Language Pre-Training, ICCV, 2023

[2] Perplexed by Perplexity: Perplexity-Based Data Pruning With Small Reference Models, ICLR, 2025

**Other Strengths And Weaknesses:**

Strength
- The proposed method is novel, simple, and effective
- Extensive experiments support the empirical strength of the proposed algorithm
- The paper is well-written and easy-to-follow

Weakness
- The reviewer's main concern is the missing of the several important works (in review and experiments as baselines)

**Questions For Authors:**

See other review sections above

**Relation To Broader Scientific Literature:**

The paper successfully extends the data pruning framework with KD, and its empirical strength promises to offer new perspectives and directions in data pruning research. Most importantly, a major strength lies in its ease of implementation.

**Theoretical Claims:**

The theoretical motivation in Section 3.3 explains the benefits of using self-distillation for enhancing training on pruned data. The authors show that self-distillation using a teacher trained on a larger dataset can reduce the bias of the student's estimation error using the context of regularized linear regression.

---

> ### Author Rebuttal · Authors · 2025-03-31
>
> We sincerely thank **Reviewer RfVw** for their careful reading, thoughtful remarks and the positive feedback. In addition, we appreciate their acknowledgement of the novelty, simplicity and effectiveness of our proposed method.
>
> Below, we kindly address each of the reviewer's questions and concerns:
>
> &nbsp;
>
> **Comparison with More Recent Approaches**
>
> We appreciate the reviewer's suggestion to compare our proposed method with more recent data pruning approaches. Following their suggestion and given the time constraints, we have recently conducted additional experiments to compare our method on CIFAR-100 with 3 out of the 4 methods proposed by the reviewer, namely:
> - **Moderate-DS [1]**
> - **D2 [2]**
> - **DUAL [3]**
>
> For a fair comparison of these methods with our method, we have utilized the official implementation of each one to generate the pruning scores using a common ResNet34 architecture. In addition, all of the necessary hyper-parameters were taken from the respective paper and/or supplementary material of each method.
>
> The results of this experiment can be viewed in the following link:
>
> [experiments_recent_pruning_methods.png](https://ibb.co/xKB0F9hq).
>
> As can observed, all pruning methods (including these recent ones) benefit from the incorporation of KD in the training process. In addition, it can be seen that on low pruning fractions random pruning + KD maintains its edge over all other methods, except for D2 + KD which achieves a similar performance. However, contrary to D2 which requires careful tuning of several hyperparameters (k,​$\gamma_{r}$,β) for each pruning fraction and each dataset, our method (simple random pruning + KD) has no such requirements and is hence more practical and easier to adopt for real-world use cases.
>
> We note that the last method the reviewer has referred to **[4]** is focused on the problem of *data pruning with re-labeling*. This sub-task of data pruning attempts to re-label noisy samples in a given dataset, and then select the subset of the data with the most accurate re-labeling of erroneous labels. Hence, all experiments in that work are strictly conducted on noisy variants of traditional datasets (e.g., CIFAR-10N, CIFAR-100N), which makes comparisons of that method with traditional data pruning approaches non-trivial.
>
> Finally, we note that we have incorporated all pruning methods suggested by the reviewer into the Related Works section.
>
> &nbsp;
>
>
> **Discussion: Application of Our Method to LLMs and VLMs**
>
> We thank the reviewer for this keen observation. For simplicity, all experiments in our paper were conducted on the image classification task in the vision domain. However, we agree that our proposed method (utilizing KD with data pruning) has great potential to be utilized in other domains, like NLP or multimodal domains. This is especially true nowadays since with the recent rise of AI-based methods over the last few years, and the high training costs that came along with them, developing more efficient training algorithms have become more important than ever. For example, we believe that it's worth exploring whether one can reduce the training costs of a given LLM (while maintaining a certain level of accuracy), by training it on a carefully pruned corpus of data with additional guidance from the logits of an informed teacher (another LLM trained on a larger corpus). We hope our work here, both empirical and theoretical, will incentivize other researchers to explore such intriguing directions in the future.
>
>
> &nbsp;
>
>
> **[1]** Moderate Coreset: A Universal Method of Data Selection for Real-world Data-efficient Deep Learning, ICLR, 2023.
>
> **[2]** D2 Pruning: Message Passing for Balancing Diversity & Difficulty in Data Pruning, ICLR, 2024.
>
> **[3]** Lightweight Dataset Pruning without Full Training via Example Difficulty and Prediction Uncertainty, ArXiv, 2025.
>
> **[4]** Robust Data Pruning under Label Noise via Maximizing Re-labeling Accuracy, NeurIPS, 2023.

---

### Official Review · Reviewer_Kpwz · 2025-03-14

**Overall Recommendation:** 2

**Summary:**

This article explores the use of knowledge distillation (KD) to improve model performance when training on pruned datasets. The authors investigate how transferring soft predictions from a teacher model can eliminate the accuracy loss caused by aggressive data pruning.

**Claims And Evidence:**

Yes, the article's claims are well-supported by theoretical analysis and extensive empirical results. Below, I assess key claims and their corresponding evidence:

1. Self-Distillation Reduces Bias in the Student Model
Claim: Training a student with a teacher model trained on a larger dataset reduces the bias in the student’s estimation error.

Evidence: The authors provide a mathematical derivation (Theorem 3.1) showing that self-distillation decreases bias.
Empirical results (Figure 5) demonstrate that increasing the teacher’s data fraction (𝑓𝑡) consistently improves student accuracy, supporting the claim.

2. Knowledge Distillation Improves Accuracy Across Different Pruning Methods and Levels
Claim: Incorporating KD consistently improves accuracy, even when training on heavily pruned datasets.

Evidence: Figures 3 & 4 show that models trained with KD outperform those without KD across all pruning methods and datasets.
Results indicate that models trained on only 10%-50% of the data with KD achieve accuracy comparable to full-data training.
In high-pruning scenarios, KD leads to 17-22% accuracy improvements over standard training (especially on CIFAR-100 and SVHN).

3. Random Pruning + KD Matches or Outperforms Sophisticated Pruning Methods
Claim: Using KD with simple random pruning can achieve accuracy comparable to or better than advanced score-based pruning methods.

Evidence: Figure 4 shows that, at high pruning levels, random pruning with KD outperforms sophisticated methods like GraNd, EL2N, and Forgetting. The authors argue that aggressive pruning can accidentally retain noisy samples, making structured pruning less effective than random selection when combined with KD.

4. The Optimal KD Weight (𝛼) depends on the Pruning Level
Claim: The effectiveness of KD depends on the balance between the pruning factor (𝑓) and KD weight (𝛼).

Evidence: Figure 6 shows that at lower pruning fractions (𝑓≤0.1), higher 𝛼 values yield better accuracy.
The authors explain that strong pruning increases label noise, and relying more on teacher predictions (higher 𝛼) helps reducing this.
As pruning becomes less aggressive, lower 𝛼 values work better since the dataset becomes cleaner.

**Essential References Not Discussed:**

NA

**Experimental Designs Or Analyses:**

The experimental design is comprehensive, evaluating knowledge distillation (KD) under different pruning levels, datasets, and teacher-student configurations. The use of multiple pruning methods (e.g., forgetting, GraNd, EL2N, and random pruning) and four datasets (CIFAR-10, CIFAR-100, SVHN, and ImageNet) strengthens the validity of the results.

However, some concerns remain:

1. The study shows that random pruning can outperform more sophisticated score-based methods at high pruning levels, but it does not deeply analyze why this occurs. Additional analysis on the types of samples retained by each method is required.

2. The study tests multiple temperature values but it does not thoroughly analyze the trade-offs between different temperatures in varying pruning conditions.

3. The claim that high-capacity teachers harm students at low pruning factors is interesting but would benefit from an ablation study isolating architecture size from other factors (e.g., regularization or training dynamics).

4. Table 1 does not contain any recent SOTA pruning method. The recent-most method compared with is from 2019.

**Methods And Evaluation Criteria:**

The methods and evaluation criteria used in the paper are appropriate for the problem of training models on pruned datasets with knowledge distillation (KD).

**Other Comments Or Suggestions:**

1. The diagram in fig. 1 is over simplistic for an ICML submission.

2. The capacity gap problem is an interesting finding, but further analysis (e.g., on feature representations or optimization dynamics) could provide deeper insights into why larger teachers degrade performance in extreme pruning scenarios.

3. The choice of KD temperature significantly impacts results, yet details on tuning and sensitivity analysis are limited.

4. Suggest experimenting with some DNNs beyond the ResNet and Wide ResNet family.

**Other Strengths And Weaknesses:**

Strengths:
The paper presents a well-motivated exploration of knowledge distillation (KD) in pruned data settings, providing both theoretical and empirical support for its claims. The experimental results are thorough, spanning multiple datasets, pruning strategies, and teacher-student configurations.

Weaknesses:

1. Theoretical claims, while sound, rely on assumptions (e.g., Gaussian noise, linear regression framework) that may not generalize to more complex deep learning models.

2. I can infer that some empirical findings, particularly regarding teacher capacity, lack deeper analysis on why larger teachers degrade performance under extreme pruning.

3. The article could improve clarity by providing more intuition behind key mathematical results, and certain methodological details (e.g., hyperparameter tuning of KD temperature) remain underexplored.

**Questions For Authors:**

1. Given that pruning methods tend to retain hard (and potentially mislabeled) samples, have you analyzed how knowledge distillation interacts with label noise? Could a poorly trained teacher amplify errors instead of eliminating them?

2. Your experiments focus on image classification tasks. Do you expect similar improvements in other domains (e.g., NLP, structured data)? Have you tested the method on regression or reinforcement learning tasks?

3. Given the computational cost of training multiple models (teacher and student) on different dataset fractions, how does this approach compare in terms of efficiency versus direct training on full or pruned datasets? Have you considered any strategies to reduce the added computational burden in real-world applications?

4. Discuss about the limitations of this method and future works to be done.

**Relation To Broader Scientific Literature:**

The paper builds on prior work in knowledge distillation (KD) and data pruning, particularly studies on model compression and sample selection.

**Theoretical Claims:**

The article presents a theoretical foundation for self-distillation in pruned data settings, but some claims lack direct empirical validation and broader generalization. While Theorem 3.1 suggests that training a teacher on a larger dataset reduces estimation bias, its proof is moved to the supplementary.

Additionally, the assumption of Gaussian noise and independent samples may not hold in real-world data distributions. The claim that KD significantly enhances generalization in low-data regimes is well-supported by empirical results but lacks a novel theoretical derivation beyond prior work. Providing explicit bias-variance decomposition experiments and relaxing assumptions in the theoretical analysis would be needed to verify the paper’s contributions.

---

> ### Author Rebuttal · Authors · 2025-03-31
>
> We thank the reviewer for the comprehensive and constructive feedback and for highlighting the strengths of our paper. We appreciate the insightful questions and will make efforts to address the reviewer’s concerns.
>
> &nbsp;
> **Incorporating Recent Pruning Methods**
>
> Following the reviewer’s suggestion, we have included three recent data pruning methods:
>
> * Moderate Coreset: A Universal Method of Data Selection for Real-world Data-efficient Deep Learning (ICLR, 2023)
>
> * D² Pruning: Message Passing for Balancing Diversity & Difficulty in Data Pruning (ICLR, 2024)
>
> * Lightweight Dataset Pruning without Full Training via Example Difficulty and Prediction Uncertainty (ArXiv, 2025)
>
>
> Accuracy results are available at the following link:
>
> [Recent_pruning_methods](https://ibb.co/xKB0F9hq)
>
> &nbsp;
> As observed, incorporating KD during training with data pruning also improves accuracy in these three pruning approaches. Also, please note that GraNd and EL2N were published in 2021, while memorization was [first applied](https://arxiv.org/abs/2206.14486) to data pruning in 2022.
>
> We are happy to include these three additional baselines in the main paper (please see our response to Reviewer RfVw).
>
> &nbsp;
> **On the Use of Simplified Assumptions in Theoretical Analysis**
>
> We acknowledge that the assumptions made in our theoretical section (Gaussian noise, independent samples) may appear limiting when considering complex real-world scenarios. However, our goal was to provide a clear and tractable analysis that offers foundational insights into the behavior of self-distillation in pruned data settings. Employing simplified assumptions, such as linear regression frameworks, is a common practice in theoretical investigations within machine learning, enabling the derivation of interpretable results. For instance:
>
> * Understanding the Gains from Repeated Self-Distillation (NeurIPS 2024)
> * A Theoretical Analysis of Fine-tuning with Linear Teachers (NeurIPS 2021)
> * Towards Data-Algorithm Dependent Generalization: a Case Study on Overparameterized Linear Regression (NeurIPS 2023).
>
> Following the reviewer's suggestion, we will include a discussion on these methodological limitations and potential extensions in a dedicated section on limitations and future work.
> Also, please note that the proof for Theorem 3.1 was moved into the supplementary due to lack of space.
>
> &nbsp;
> **On Why Larger Teachers Degrade Performance under Extreme Pruning and Impact of KD Temperature**
>
> We thank the reviewer for the opportunity to clarify this observation. Below, we provide our intuition based on our current understanding.
> We hypothesize that learning complex decision boundaries becomes increasingly challenging when the number of samples is significantly reduced. Consequently, when training a student model on a pruned dataset with a high pruning ratio, smaller teachers tend to be more beneficial. Larger teachers, while capable of handling hard samples, can cause the student to focus on these difficult instances, which are inherently hard to resolve given the limited data. In contrast, smaller teachers are more tolerant to errors from hard samples and guide the student toward more manageable patterns.
> While increasing the temperature can soften the teacher's output, the softmax temperature is a global transformation that affects all logits simultaneously, and thus may not sufficiently reduce the impact of complex decision boundaries imposed by larger teachers.
>
> We illustrate this in the following figure:
>
> [Teacher_capactity_accumulated_predictions.png](https://ibb.co/WNGK191R)
>
> &nbsp;
> **Details on Tuning KD Temperature and Sensitivity Analysis**
>
> Following the reviewer's suggestion to analyze the KD temperature, we kindly refer to Figure 10 in the appendix, where we present accuracy results for various temperature values across different pruning fractions and architectures.
>
> &nbsp;
> **Additional Questions**
>
> 1. Pruning methods can retain hard or noisy samples, leading to poor student performance (see Section 3.2). However, training a teacher on a (noisy) pruned dataset is not recommended, as it can result in a poorly trained teacher that degrades student’s performance.
> 2. We expect our approach to generalize to other domains. Data pruning with KD is promising for NLP tasks, and while not tested on regression, we believe it could be suitable. As for RL, further investigation is needed.
> 3. Data pruning reduces computational burden, e.g., in HPO or Active Learning, training many models on a small portion of the data with soft predictions from a full-data-trained teacher maintains high accuracy while lowering computational costs.
>
> 4. As suggested, we will include a section on the method's limitations and future work.
>
> Please let us know if you'd like us to elaborate on any point, as this grants 5,000 more characters.
>
> &nbsp;
> We hope our response clarifies your concerns, and we would appreciate it if you could consider increasing your score.

---

### Decision · Program_Chairs · 2025-05-01

**Decision:**

Accept (poster)

**Comment:**

Three of the four reviewers rated the paper positively (1 accept, 2 weak accept, one weak reject).

The paper presents a novel approach to dataset pruning, which augments the training process with soft labels from a teacher network that was trained on the full dataset.

The paper's claims are well-supported by theoretical analysis and extensive empirical results. In particular, the main claim of this paper - that a simple random pruning method, when combined with knowledge distillation (KD), outperforms all standard data pruning algorithms that rely on hard labels - is supported by clear and convincing evidence.

There was a concern about the novelty of combining knowledge distillation with data pruning. However the review discussion clarified that while there is some recent work on dataset distillation, the novelty of the paper lies in utilizing knowledge distillation within data pruning.
The paper demonstrates that in the presence of KD, model accuracy remains robust regardless of the data pruning method employed, suggesting that simple random pruning, when combined with KD, is a viable alternative to more sophisticated pruning methods.

The assumption of Gaussian noise and independent samples may not hold in real-world data distributions. The claim that KD significantly enhances generalization in low-data regimes is well-supported by empirical results but lacks a novel theoretical derivation beyond prior work. Explicit bias-variance decomposition experiments and relaxing assumptions in the theoretical analysis would strengthen the paper.

Two of the reviewers suggest examining more closely the role of temperature. The study tests multiple temperature values but it does not thoroughly analyze the trade-offs between different temperatures in varying pruning conditions. The choice of KD temperature significantly impacts results, yet details on tuning and sensitivity analysis are limited. However, as pointed out by the authors, this issue is covered already to some extent in appendix D, figure 10, which should be merged with the main paper.